# Effects of air pollution control policies on PM$_{2.5}$ pollution improvement in China from 2005 to 2017: a satellite based perspective

Zongwei Ma [1, 2], Riyang Liu [1], Yang Liu [3], Jun Bi [1, 2, *]

[1] State Key Laboratory of Pollution Control and Resource Reuse, School of the Environment, Nanjing University, Nanjing, Jiangsu, China

[2] Jiangsu Collaborative Innovation Center of Atmospheric Environment and Equipment Technology (CICAEET), Nanjing University of Information Science & Technology, Nanjing, Jiangsu, China

[3] Department of Environmental Health, Rollins School of Public Health, Emory University, Atlanta, GA, USA

**\* Correspondence to:**

**Dr. Jun Bi**, School of the Environment, Nanjing University, 163 Xianlin Avenue, Nanjing 210023, P. R. China. Tel.: +86 25 89681605. E–mail address: jbi@nju.edu.cn.

# ABSTRACT

Understanding the effectiveness of air pollution control policies is important for future policy making. China implemented strict air pollution control policies since 11th Five Year Plan (FYP). There is still a lack of overall evaluation of the effects of air pollution control policies on $PM_{2.5}$ pollution improvement in China since 11th FYP. In this study, we aimed to assess the effects of air pollution control policies from 2005 to 2017 on $PM_{2.5}$ from the view of satellite remote sensing. We used the satellite derived $PM_{2.5}$ of 2005-2013 from one of our previous studies. For the data of 2014-2017, we developed a two-stage statistical model to retrieve satellite $PM_{2.5}$ data using the Moderate Resolution Imaging Spectroradiometer (MODIS) Collection 6 aerosol optical depth (AOD), assimilated meteorology, and land use data. The first-stage is a day-specific linear mixed effect (LME) model and second-stage is generalized additive model (GAM). Results show that the Energy Conservation and Emissions Reduction (ECER) policy, implemented in 11th FYP period and focused on $SO_2$ emissions control, had co-benefits on $PM_{2.5}$ reductions. The increasing trends of $PM_{2.5}$ pollution (1.88 and 3.14 $\mu g/m^3$/year for entire China and Jingjinji Region in 2004-2007, $p<0.005$) was suppressed after 2007. The overall $PM_{2.5}$ trend for entire China was -0.56 $\mu g/m^3$/year with marginal significance ($p=0.053$) and $PM_{2.5}$ concentrations in Pearl River Delta Region had a big drop (-4.81 $\mu g/m^3$/year, $p<0.001$) in 2007-2010. The ECER policy during 12th FYP period were basically the extension of 11th FYP policy. $PM_{2.5}$ is a kind of composite pollutant which comprises primary particles and secondary particles such as sulfate, nitrate, ammonium, organic carbon, elemental carbon, etc. Since ECER policy focused on single-pollutant control, it had shown great limitation for $PM_{2.5}$ reductions. The $PM_{2.5}$ concentrations did not decrease from 2010 to 2013 in polluted areas ($p$ values of the trends were greater than 0.05). Therefore, China implemented two stricter policies: 12th FYP on Air Pollution Prevention and Control in Key Regions (APPC-KR) in 2012, and Action Plan of Air Pollution Prevention and Control (APPC-AP) in 2013. The goal of air quality improvement (especially $PM_{2.5}$ concentration improvement) and measures for multi-pollutant control were proposed. These policies had led to dramatic decrease in $PM_{2.5}$ after 2013 (-4.27 $\mu g/m^3$/year for entire China in 2013-2017, $p<0.001$).

## 1 Introduction

Fine particulate matter ($PM_{2.5}$, particulate matter with aerodynamic diameter less than 2.5 μm) is a major atmospheric pollutant, which has been shown to be strongly associated with adverse health effects (e.g., cardiovascular and respiratory morbidity and mortality) in many epidemiological studies (Crouse et al., 2012;Dominici et al., 2006;Pope et al., 2002). With the rapid economic development and industrialization in the past decades, $PM_{2.5}$ pollution has gradually become a major environmental issue in China (Liu et al., 2017a). However, the Chinese government did not focus on the $PM_{2.5}$ issues until 2012. Therefore, air pollution control policies implemented before 2012 mainly focus on $SO_2$, industrial dust and soot emission control. The air pollution control policies of China started to pay attention to $PM_{2.5}$ since late 2012.

Understanding the effectiveness of air pollution controls policies is important for future air pollution control in China. Several studies have examined the historical air pollution control policies and their association with the trends of $SO_2$, $NO_2$, and $PM_{10}$ (Jin et al., 2016;Chen et al., 2011;Hu et al., 2010). Since the national $PM_{2.5}$ monitoring network was established in late 2012, few studies have evaluated the effects of air pollution control policies on $PM_{2.5}$ concentrations before 2013 due to the lack of historical ground monitoring data. Therefore, it is difficult to understand whether the air pollution control policies had synergistic effects on $PM_{2.5}$ reductions before 2012.

In recent years, many studies have shown that satellite remote sensing provides a powerful tool to assess the spatiotemporal trends of air pollutions for both global and regional scales (Miyazaki et al., 2017;Itahashi et al., 2012;Krotkov et al., 2016). Estimating ground $PM_{2.5}$ using satellite aerosol optical depth (AOD) data was also an effective way to fill the spatiotemporal $PM_{2.5}$ gaps left by ground monitoring network (Liu, 2013, 2014;Hoff and Christopher, 2009). There are two major methods to estimate ground $PM_{2.5}$ concentration using AOD data, i.e., the scaling method and statistical approach (Liu, 2014). The scaling method uses atmospheric chemistry models to simulate the association between AOD and $PM_{2.5}$, and then calculate the satellite-derived $PM_{2.5}$ using the

equation: $Satellite\text{-}derived\ PM_{2.5} = \frac{Simulated\ PM_{2.5}}{Simulated\ AOD} \times Satellite\ AOD$ (Liu, 2014). Boys et al.

(2014) and van Donkelaar et al. (2015) estimated the global satellite $PM_{2.5}$ time series using the

scaling method. Compared to the scaling method, statistical models have greater prediction

accuracy but require large amount ground-measured $PM_{2.5}$ data to develop the models (Liu, 2014).

By taking advanced of the newly established ground $PM_{2.5}$ monitoring network, we developed a

two-stage statistical model to estimate historical monthly mean $PM_{2.5}$ concentrations using Aqua

Moderate Resolution Imaging Spectroradiometer (MODIS) Collection 6 AOD data in one of our

previous studies (Ma et al., 2016). Validation results shows that this monthly $PM_{2.5}$ dataset has high

prediction accuracy ($R^2 = 0.73$). This accurate historical $PM_{2.5}$ dataset from 2004 to 2013 allowed us

to examined the effects of pollution control policies on $PM_{2.5}$ concentrations. In this previous

study(Ma et al., 2016), we preliminarily analyzed the effects of Energy Conservation and Emissions

Reduction (ECER) policy in 11[th] five year plan (2006-2010). We found an inflection point around

2008, after which $PM_{2.5}$ concentration showed slight decreasing trend, showing the co-benefits of

the ECER policy. From 2013 to 2017, China implemented the Action Plan of Air Pollution

Prevention and Control (APPC-AP), which focused on $PM_{2.5}$ pollution. Currently, there is still a

lack of overall evaluation of the effects of air pollution control policies on $PM_{2.5}$ pollution

improvement in China from 2005 to 2017.

      In this study, we aimed to assess the effects of air pollution control policies from 2005 to 2017

on $PM_{2.5}$ from the view of satellite remote sensing. We used the satellite-derived $PM_{2.5}$ dataset

developed in our previous study (Ma et al., 2016). Since this dataset was from 2004 to 2013 and

data after 2014 has been lacking, we extended the dataset to 2017 in the present work. To keep

consistent with our previous satellite $PM_{2.5}$ dataset, we used the same method as described in our

previous study (Ma et al., 2016).

**2 Overview of air pollution control policies in China from 2005 to 2017**

During 2005 to 2017, China implemented a series air pollution prevention and control policies,

including 11[th] Five Year Plan (FYP) on Environmental Protection (2006-2010), ECER Policy

during 11[th] FYP period, 12[th] FYP on Environmental Protection (2011-2015), 12[th] FYP on ECER,

The 12[th] FYP on Air Pollution Prevention and Control in Key Regions (APPC-KR), and APPC-AP

(2013-2017). The base year, implementation period, major goals, and major measures are listed in

Table 1.

During 11[th] FYP period, there was no specific air pollution control policy. Air pollution

prevention and control measures were incorporated in the whole environmental protection plan or

policy (i.e., 11[th] FYP on Environmental Protection and ECER policy). From Table 1 we can see that

the air pollution policies during 11[th] FYP mainly focused on total emission reduction. In this period,

environmental management in China was emission control oriented, that is, the indicators for local

governments' environmental performance assessment were emission reduction rates, not the

environmental quality. The 12[th] FYP on Environmental Protection and ECER policy were basically

the extension of the 11[th] FYP policies, which mainly focused on emission reduction.

       The 12[th] FYP on APPC-KR is the first special plan for air pollution prevention and control.

This plan proposed the idea of unification of total emission reduction and air quality improvement.

And it proposed the goals of air pollutant concentration control for the first time. $PM_{2.5}$ pollution

control was also incorporated in this plan. Although the implementation period of 12[th] FYP on

APPC-KR is 2011-2015, it was issued in October 29, 2012. After that, China issued the APPC-AP

(2013-2017) in September 10, 2013, which strengthened the air pollution control and the goals of

air quality improvement. These policies indicated that the focus of air pollution control in China

began to focus on $PM_{2.5}$ concentrations reductions.

## 3 Data and method

### 3.1 Satellite-based $PM_{2.5}$ from 2004 to 2013

       We estimated the monthly satellite-based $PM_{2.5}$ data from 2004 to 2013 at 0.1 °resolution in

our previous work (Ma et al., 2016). Briefly, we developed a two-stage statistical model using

MODIS Collection 6 AOD and assimilated meteorology, land use data, and ground monitored $PM_{2.5}$

concentrations in 2013. The overall model cross-validation $R^2$ (coefficient of determination) was

0.79 (daily estimates) for the model year. Since ground monitor data before 2013 has been lacking and therefore it is unable to develop statistical models before 2013 to estimate historical $PM_{2.5}$ concentrations. Thus, the historical $PM_{2.5}$ concentrations (2004-2012) were then estimated using the model developed based on 2013 model. Two ways were used to validate the accuracy of historical estimates. First, we compared the historical estimates monitoring data from Hong Kong and Taiwan before 2013. Second, we estimated $PM_{2.5}$ concentrations in the first half of 2014 using the 2013 model and compared them with the ground measurements to evaluate the accuracy of $PM_{2.5}$ estimates beyond the model year, which can represent the accuracy of historical estimates. Validation results indicated that it accurately predicted $PM_{2.5}$ concentrations with little bias at the monthly level ($R^2 = 0.73$, slope $= 0.91$).

For $PM_{2.5}$ concentrations from 2004 to 2013, we directly used above-mentioned satellite-based $PM_{2.5}$ dataset, which was estimated using the model developed in 2013. First, this dataset has been shown high accuracy and has been widely used in environmental epidemiological (Liu et al., 2016a;Wang et al., 2018a), health impact (Liu et al., 2017b;Wang et al., 2018b), and social economic impact (Chen and Jin, 2019;Yang and Zhang, 2018) studies in China. Second, a recent study has shown that the historical hindcast ability of the annual model decreased when hindcast year was long before the model year (Xiao et al., 2018). Therefore, we did not use the models of 2014 to 2017 to estimate the hindcast $PM_{2.5}$.

**3.2 Satellite-based $PM_{2.5}$ from 2014 to 2017**

Unlike historical estimates from 2004 to 2012, we have sufficient ground monitored $PM_{2.5}$ data to develop statistical models after 2013, which allowed us to estimate daily $PM_{2.5}$ concentrations accurately. Therefore, we developed a separate $PM_{2.5}$-AOD statistical model for each year of 2014-2017 to estimate the spatially-resolved (0.1 °resolution) $PM_{2.5}$ concentrations. To keep satellite $PM_{2.5}$ estimates of 2014-2017 consistent with our previous satellite $PM_{2.5}$ dataset, we used the same method as described in our previous study (Ma et al., 2016). The data, model development, and model validation are briefly introduced as follows.

The data used in this study include ground monitored $PM_{2.5}$ concentrations ($\mu g/m^3$), Aqua

MODIS Collection 6 Dark Target (DT) AOD and Deep Blue (DB) AOD data, planetary boundary

layer height (PBLH, 100 m), wind speed (WS, m/s) at 10 m above the ground, mean relative

humidity in PBL (RH_PBLH, %), surface pressure (PS, hPa), precipitation of the previous day

(Precip_Lag1; mm), MODIS active fire spots, urban cover (%), and forest cover (%). Ground

monitored $PM_{2.5}$ data were collected from China Environmental Monitoring Center (CEMC),

environmental protection agencies of Hong Kong and Taiwan. Figure 1 shows the ground $PM_{2.5}$

monitors used in this study. AOD were downloaded from the Level 1 and Atmospheric Archive and

Distribution System (https://ladsweb.modaps.eosdis.nasa.gov/, accessed on Mar 29, 2019).

Meteorological data were extracted from Goddard Earth Observing System Data Assimilation

System GEOS-5 Forward Processing (GEOS 5-FP) meteorological data (ftp://rain.ucis.dal.ca,

accessed on Mar 29, 2019). MODIS fire spots were from the NASA Fire Information for Resource

Management System (https://earthdata.nasa.gov/earth-observation-data/near-real-time/firms,

accessed on Mar 29, 2019). Land use information were downloaded from Resource and

Environment Data Cloud Platform of Chinese Academy of Science

(http://www.resdc.cn/data.aspx?DATAID=184, accessed on Mar 29, 2019).

Previous studies have shown the data quality issue of ground $PM_{2.5}$ measurements from CEMC

network (Liu et al., 2016b;Rohde and Muller, 2015). We performed the data screening procedure

before model fitting. Abnormal values (extreme high or extreme low values for a site compared

with its neighboring sites, repeated values for continuous hours, etc.) were deleted before model

fitting. We required at least 20 hourly records to calculate the daily average $PM_{2.5}$ concentrations.

DT and DB AOD were combined using inverse variance weighting method to improve the spatial

coverage of AOD data (Ma et al., 2016). This combined AOD data has been shown good

consistency ($R^2$=0.8, mean bias=0.07) with ground AOD measurements from Aerosol Robotic

Network (AERONET) (Ma et al., 2016). All data were assigned to a predefined 0.1 °grid. Then all

of the variables were matched by grid cell and day-of-year (DOY) for model fitting.

A two-stage statistical model was developed for each year separately from 2014 to 2017. The

first-stage linear mixed-effects (LME) model included day-specific random intercepts and slopes for AOD, season-specific random slopes for meteorological variables, and fixed slope for precipitation and fire spots. The model structure of first-stage model was shown as follows:

$PM_{2.5,st} = (\mu + \mu') + (\beta_1 + \beta_1')AOD_{st} + (\beta_2 + \beta_2')WS_{st} + (\beta_3 + \beta_3')PBLH_{st} + (\beta_4 + \beta_4')PS_{st} + (\beta_5 +$

$\beta_5')RH\_PBLH_{st} + \beta_6Precip\_Lag1_{st} + \beta_7Fire\_spots_{st} + \varepsilon_{1,st}(\mu'\beta_1') \sim N[(0,0), \Psi_1] +$

$\varepsilon_{2,sj}(\beta_2'\beta_3'\beta_4'\beta_5') \sim N[(0,0,0,0), \Psi_2]$  (1)

where $PM_{2.5,st}$ is ground $PM_{2.5}$ measurements at grid cell $s$ on DOY $t$; $AOD_{st}$ is DT-DB merged AOD; $WS_{st}$, $PBLH_{st}$, $PS_{st}$, $RH\_PBLH_{st}$, $Precip\_Lag1_{st}$ are meteorological variables; $Fire\_spots_{st}$ is the fire count; $\mu$ and $\mu'$ are the fixed and day-specific random intercepts, respectively; $\beta_1$-$\beta_7$ are fixed slopes; $\beta_1'$ is the day-specific random slope for AOD; $\beta_2'$-$\beta_5'$ are the season-specific random slopes for meteorological variables; $\varepsilon_{1,st}$ is the error term at grid cell $s$ on DOY $t$; $\varepsilon_{2,sj}$ is the error term at grid cell $s$ in season $j$; $\Psi_1$ and $\Psi_2$ are the variance-covariance matrices for the day- and season-specific random effects, respectively. The first-stage model was fitted for each province separately. We created a buffer zone for each province to include data with at least 3,000 data records and at least 300 days. We averaged overlapped predictions from neighboring provinces to generate a smooth national $PM_{2.5}$ concentration surface.

The second-stage generalized additive model (GAM) established the relationship between the residuals of the first-stage model and smooth terms of geographical coordinates, forest and urban cover.

$PM_{2.5}\_resid_{st} = \mu_0 + s(X, Y)_s + s(ForestCover)_s + s(UrbanCover)_s + \varepsilon_{st}$  (2)

where $PM_{2.5}\_resid_{st}$ is the residual of first-stage model at grid cell $s$ on DOY $t$; $\mu_0$ is the intercept; $s(X, Y)_s$ is the smooth term of the coordinates of the centroid of grid cell $s$; $s(ForestCover)_s$ and $s(UrbanCover)_s$ are the smooth functions of forest cover and urban area for grid cell $s$; and $\varepsilon_{st}$ is the error term.

10-fold cross validation (CV) was used to evaluate the model over-fitting, that is, the model could have better prediction performance in the model fitting dataset than the data which are not

included model fitting. In 10-fold CV, all samples in the model dataset are randomly and equally divided into ten subsets. One subset was used as testing samples and the rest subsets are used to fit the model. This process was repeated for 10 rounds until each subset was used for testing for once. Statistical indicators of coefficient of determination ($R^2$), mean prediction error (MPE), and root mean squared prediction error (RMSE) were calculated and compared between model fitting and CV to assess model performance and over-fitting.

### 3.3 Time series analysis

Monthly mean $PM_{2.5}$ concentrations for each grid cell were calculated to perform the time series analysis. Following our previous study (Ma et al., 2016), we required at least six daily $PM_{2.5}$ predictions in each month to calculate the monthly mean $PM_{2.5}$. We deseasonalized the monthly $PM_{2.5}$ time series by calculating the monthly $PM_{2.5}$ anomaly time series for each grid cell to remove the seasonal effect. $PM_{2.5}$ trend for each grid cell was calculated using least squares regression (Weatherhead et al., 1998):

$$(PM_{2.5})_{anomaly, \, s, \, m} = (PM_{2.5})_{s, \, m} - \overline{(PM_{2.5})_{s, \, j}} \qquad m = 1, 2, 3, ..., M \qquad j = 1, 2, 3, ..., 12 \qquad (3)$$

$$(PM_{2.5})_{anomaly, \, s, \, m} = \mu + \beta \times m + \varepsilon, \qquad m = 1, 2, 3, ..., M \qquad (4)$$

where $(PM_{2.5})_{anomaly, \, s, \, m}$ is the $PM_{2.5}$ anomaly at grid cell $s$ for month $m$ during the calculating period; $(PM_{2.5})_{s, \, m}$ is the estimated $PM_{2.5}$ concentration at grid cell $s$ for month $m$; $m$ is the month index and $M$ is the total number of months during the calculating period (2004-2017, $M=168$); $\overline{(PM_{2.5})_{s, \, j}}$ is the 14-year average $PM_{2.5}$ concentration of the month to which month $m$ belongs ($j = 1$ for January, $j = 2$ for February, ⋯, etc.); $\mu$ is the intercept; $\beta$ is the slope, which is also the trend of $PM_{2.5}$ (μg/m³/month); $\varepsilon$ is the error term. The annual $PM_{2.5}$ trend (μg/m³/year) = $12 \times \beta$. The method of $t$ test was used to obtain the statistical significance of the trends. This method has been successfully applied to trend analyses of monthly mean $PM_{2.5}$ and AOD anomaly time-series data (Hsu et al., 2012;Boys et al., 2014;Zhang and Reid, 2010;Xue et al., 2019).We analyzed the $PM_{2.5}$ trend for different periods to examine the effects of air pollution control policies on $PM_{2.5}$ pollution improvement.

## 4. Results and discussion

### 4.1 Validation of satellite-based PM$_{2.5}$ concentrations from 2014 to 2017

Table S1 (Supplemental Materials, SM) summarized the statistics of all variables for the modeling dataset from 2014 to 2017. Overall, there are 95 649, 110 805, 113 490, and 123 652 matchups for the model fitting datasets for years of 2014, 2015, 2016, and 2017, respectively. The average PM$_{2.5}$ concentration decreases year by year, from 65.66 μg/m$^3$ in 2014 to 48.32 μg/m$^3$ in 2017. Correspondingly, the average AOD also shows a decreasing trend from 0.67 in 2014 to 0.50 in 2017.

Figure 2 shows the model fitting and cross validation results for each year's model. The model fitting R$^2$ ranges from 0.75 (2015) to 0.80 (2017) and CV R$^2$ ranges from 0.72 (2015) to 0.77 (2017), which is similar to the 2013 model (0.82 for model fitting and 0.79 for CV) developed in our previous study (Ma et al., 2016). The model prediction accuracy is different among years, which is consistent with previous studies. Hu et al. (2014) studied the 10-year spatial and temporal trends of PM$_{2.5}$ concentrations in the southeastern US from 2001 to 2010. They developed a separate two-stage statistical model for each year and found the CV R$^2$ ranged from 0.62 in 2009 to 0.78 in 2005 and 2006. Kloog et al. conducted two studies in Northeast US and also found that the validation R$^2$ varied among years (Kloog et al., 2011;Kloog et al., 2012). Compared to the model fitting R$^2$, the CV R$^2$ only decreases 0.02 in 2016 and 0.03 in 2014, 2015, and 2017, showing that our models were not substantially over-fitted. For the monthly mean concentrations calculated from at least six daily PM$_{2.5}$ predictions, the validation R$^2$ values ranges from 0.75 to 0.81 (Figure 3). The results show that the overall prediction accuracy of the models from 2014 to 2017 is satisfying.

The fixed effects, model fitting, and CV results of the first-stage LME model for each province are shown in Tables S2-S5 (SM). AOD is the only variable that was statistically significant in all provincial models for all years ($p < 0.05$). Wind speed, relative humidity, precipitation, and fire spots were significant in most provincial models. The CV R$^2$ varies for different province and different year. The CV R$^2$ values range from 0.61 in Xinjiang to 0.77 in Heilongjiang for 2014,

from 0.34 in Xinjiang to 0.76 in Hebei for 2015, from 0.44 in Tibet to 0.77 in Jiangsu for 2016, and from 0.38 in Xinjiang to 0.79 in Sichuang for 2017. We also fitted a first-stage LME model for entire China. Results show that the overall CV $R^2$ values for first-stage LME model dropped to 0.57, 0.52, 0.56, and 0.54, for 2014, 2015, 2016, and 2017, respectively. Therefore, fitting the first-stage model for each province separately can greatly improve the prediction accuracy.

A potential source of uncertainties of statistical models is the uneven spatial distribution of ground PM$_{2.5}$ monitors. The CEMC air quality network mainly covers large urban centers with very limited sites coverage in rural areas, especially in western part of the country. Since it requires large amount ground-measured PM$_{2.5}$ data to develop satellite-based statistical model, this bias cannot be avoided. Despite this limitation, high model performances have been achieved in this study and previous similar studies (Zheng et al., 2016;Huang et al., 2018;Xue et al., 2019), which are much better than the scaling method. For example, Geng et al. (2015) estimated long-term PM$_{2.5}$ concentrations in China using scaling method and found the validation $R^2$ of PM$_{2.5}$ predictions was 0.72 compared to the five-month averaged ground PM$_{2.5}$ concentrations for Jan-May, 2013. A global study of PM$_{2.5}$ estimates combing scaling and statistical methods shows that their validation $R^2$ of long-term average PM$_{2.5}$ was 0.67 for their first-stage scaling method (van Donkelaar et al., 2016).

**4.2 Overall spatial and temporal trend of PM$_{2.5}$ concentrations in China from 2004 to 2017**

Figure 4 shows that spatial distribution characteristics of annual mean PM$_{2.5}$ concentrations are similar among the years from 2004 to 2017. The most polluted area was North China Plain (including south of Jingjinji Region, Henan, and Shandong Provinces), which was also the largest polluted area. The Sichuan Basin (including east of Sichuan and western Chongqing) is another polluted area. The cleanest areas were mainly located in Tibet, Hainan, Taiwan, Yunnan, and the north of Inner Mongolia. The spatial distributions of satellite-derived PM$_{2.5}$ concentrations from 2013 to 2017 are consistent with the spatial characteristics of ground monitored PM$_{2.5}$ (Figure S2, SM)

Figure 5 shows the spatial distributions of $PM_{2.5}$ trends and significance levels in China from 2004 to 2017. Over all, the $PM_{2.5}$ pollution level of most area in China showed a decreasing trend ($p<0.05$). Figure 6 and Table 2 shows that the overall trends of 2004-2017 for entire China, Jingjinji, Yangtze River Delta (YRD), Pearl River Delta (PRD) Regions were -1.27, -1.55, -1.60, and -1.27 $\mu g/m^3$/year (all $p<0.001$), respectively. Back to Figure 4, we can see that the decrease of $PM_{2.5}$ mainly happened after 2013. $PM_{2.5}$ concentrations had an obvious increase from 2004 to 2007. The area with $PM_{2.5}$ concentrations higher than 100 $\mu g/m^3$ continuously expanded during this period. From 2008 to 2013, the pollution levels plateaued in most areas. After 2013, the $PM_{2.5}$ concentrations obviously decreased.

**4.3 Effect of ECER policy during 11th Five Year Plan period**

To assess the effect of ECER policy during 11th FYP, we calculated the trends of $PM_{2.5}$ for 2005-2010, 2004-2007, and 2007-2010 for each grid cell (Figure 7).

Compared to the base year (2005) of the 11th FYP period, the overall $PM_{2.5}$ pollution of 2010 did not have obvious change. Some of the area had decreasing trends (Figure 7(a)) but the trends were insignificant (Figure 7(b)). Some regions (Shandong, Henan, Jiangsu Provinces, and Northeast China) had slight increasing trend (~1-2 $\mu g/m^3$/year, $p<0.001$). Overall, the trends of entire China, Jingjinji, YRD, and PRD Regions were all insignificant (0.41, 0.26, 0.61, and -1.26 $\mu g/m^3$/year, and all $p>0.1$) during 11th FYP period.

However, when separating this period into two periods, we can see that before 2007, the $PM_{2.5}$ concentrations generally had significant increasing trends (Figure 7(c, d)), especially in South of Jingjinji Region, Henan, Shandong, and Hubei Provinces. The overall trends of entire China and Jingjinji Region are 1.88 ($p<0.001$) and 3.14 ($p<0.005$) $\mu g/m^3$/year (Table 2). The trends of YRD and PRD Regions are insignificant. During the 10th YFP period, China missed the emission control goals. The emission of sulfur dioxide increased by ~28% (Xue et al., 2014;Schreifels et al., 2012). The 11th FYP for National Economic and Social Development of China released in 2006 proposed the ECER goals. However, China did not achieve the annual goal in 2006. These could explain the

increasing trend of PM$_{2.5}$ during 2004-2007.

After that, China released the Comprehensive Working Plan on ECER (http://www.gov.cn/zwgk/2007-06/03/content_634545.htm, accessed on Mar 29, 2019) in 2007 to strengthen the ECER measures. Major control measures included (Schreifels et al., 2012):

implementing flue gas desulphurization for coal-fired power plant, closing inefficient and backward production capacity, implementing energy conservation projects, increasing pollution levy for SO$_2$ emission, recommending baghouse dust filter for industrial soot and dust emission control etc. As a result, great achievements had been made at the end of 11[th] FYP (Schreifels et al., 2012;Zhou et al., 2015): total emission of SO$_2$ decreased by ~14% compared to the level of 1995; approximate 86% of the power plant were installed with desulphurization facilities in 2010 compared to 14% in 2005; nearly 80 GW of small coal-fired power units were closed during 2006-2010; soot emission of coal-fired power plants in 2010 was reduced by 55.6% compared with that in 2005, etc.

Due to these control measures, the increasing trend of PM$_{2.5}$ pollution was suppressed after 2007. PM$_{2.5}$ concentrations of Central and South China decreased significantly, with highest trend of around -9 μg/m$^3$/year (Figure 7(e, f), $p<0.01$). For south of Jingjinji Region, Henan, Shandong, and Hubei Provinces, which had significantly increased before 2007, showed insignificant trends (Figure 7(f), $p>0.05$). Table 2 shows that the overall PM$_{2.5}$ trend for entire China was -0.56 μg/m$^3$/year with marginal significance ($p=0.053$). Overall trends of Jingjinji and YRD Regions were not significant during the latter half of 11[th] FYP period. And PM$_{2.5}$ concentrations in PRD Region had a big drop (-4.81 μg/m$^3$/year, $p<0.001$). Results show that although air pollution control policies of 11[th] FYP were not designed for PM$_{2.5}$ prevention and control, they still had co-benefits on PM$_{2.5}$ pollution control. There were two main reasons. First, SO$_2$ is the precursor gas of sulfate. Previous studies have shown that sulfate was the major component of PM$_{2.5}$ during 11[th] FYP period(Li et al., 2009;Li et al., 2010;Pathak et al., 2009). The reduction of SO$_2$ could therefore contribute to the suppression of increasing PM$_{2.5}$ pollution. Second, the control of industrial dust and soot, which include a portion of primary PM$_{2.5}$ (Yao et al., 2009), also contributed to the PM$_{2.5}$ pollution reduction.

**4.4 Effect of air pollution control policies in 12th Five-Year Plan period (2011–2015)**

Figure 8(a) and (b) show that most of the areas of China show significant decreasing trend during 12th FYP period. $PM_{2.5}$ concentrations of entire China, Jingjinji, and YRD had dropped by 2.89, 3.63, and 3.33 $\mu g/m^3$/year ($p<0.001$). When considering the years from 2010 to 2013, although overall trend of entire China was -1.03 $\mu g/m^3$/year ($p<0.05$, Table 2), the decreasing trend mainly happened in Xinjiang and Central Inner Mongolia. The deserts in Xinjiang and Inner Mongolia are the major sources of dust pollution in China. A recent study showed that dust is the largest contributor to $PM_{2.5}$ over this region (Philip et al., 2014). The change in natural dust in desert areas may be the major contributor to the decreasing trend of $PM_{2.5}$ during 2010-2013. Most of the polluted area in China did not had obvious change (Figure 8(c) and (d)). As we mentioned above, The ECER policy during 12th FYP period was basically the extension of the policy in 11th FYP, which mainly focused on emissions reduction. As the further development of social economic, the ECER policy had shown its limitation for $PM_{2.5}$ reductions. $PM_{2.5}$ is a kind of composite pollutant and its constituents includes primary particles and secondary particles such as sulfate, nitrate, ammonium, organic carbon, elemental carbon, etc. With the deepening of $SO_2$ and industrial dust/soot emission reduction, their contributions to $PM_{2.5}$ pollution control would reduce. Although 12th FYP on Environmental Protection also proposed 10% reduction of $NO_x$ from 2010 to 2015. However, along with economic growth in China, the benefits of emission control for single-pollutant could be offset by increased energy usage. Considering the complicated $PM_{2.5}$ compositions, comprehensive and coordinated control measures for multiple pollutants were urgently needed.

Therefore, China issued the 12th FYP on APPC-KR in late 2012, which is the first special plan for air pollution prevention and control and focused on air quality improvement. APPC-KR proposed a series of key projects which included 477 $SO_2$ treatment projects, 755 $NO_x$ treatment projects, 10 073 industrial soot and dust treatment projects, 1 311 VOCs treatment projects in key industrial sectors, 281 vapor recovery projects for oil and gas. 188 yellow-sticker vehicle elimination projects, 192 fugitive dust comprehensive treatment projects, and 122 capacity building

projects. An English translation version of APPC-KR and its key projects has been prepared by Clean Air Alliance of China (CAAC) and can be found elsewhere (http://www.cleanairchina.org/product/6347.html, accessed on Mar 29, 2019) (CAAC, 2013c, a).

In addition, in 2012, China issued a new air quality standard, i.e., the *National Ambient Air Quality Standard of China* (NAAQS) (GB 3095-2012). Compared with the former NAAQS (GB 3095-1996) issued in 1996, this new standard incorporated $PM_{2.5}$ as a major control pollutant. According to GB 3095-2012, the Level 1 annual mean standard of $PM_{2.5}$ is 15 μg/m$^3$, which is assigned for protecting the air quality of natural reserves and scenic areas and is equivalent to the World Health Organization (WHO) Air Quality Interim Target-3 (IT-3) Level. The Level 2 standard of 35 μg/m$^3$ is designated for residential, cultural, industrial, and commercial areas, which is equivalent to WHO Air Quality Interim Target-1 (IT-1) Level. Meanwhile, a comprehensive real-time air quality monitoring network covering 74 major Chinese cities was established in late 2012.

The implementation of APPC-KR, together with the implementation of APPC-AP starting from 2013 (shown in the following section), had led to dramatic drops in $PM_{2.5}$ concentrations in China after 2013. Table 3 shows $PM_{2.5}$ concertation improvement goals and final accomplishments for key regions (see Figure S1, SM) of 12[th] FYP on APPC-KR calculated from satellite $PM_{2.5}$. Results show that all key regions had accomplished the goals except for Yinchuan. The changes in population weighted averages also show similar results. Overall, the 12[th] FYP on APPC-KR accomplished its air pollution control goals. And the decrease of $PM_{2.5}$ concentrations was mainly attributable to the decrease after 2013.

**4.5 Effect of Action Plan for Air Pollution Prevention and Control (2013-2017)**

China issued the APPC-AP (2013-2017) in late 2013, which further strengthened the air pollution control measures and air quality improvement goals. The air pollution control measures included ten categories:

- Increase effort for comprehensive pollution control, reduce emissions of multi-pollutants;
- Optimize industrial structure, promote industrial restructuring;

- Accelerate technology transformation, improve innovation capability;

- Adjust energy structure, increase clean energy supply;

- Strengthen environmental thresholds, optimize industrial layout;

- Promote the role of market mechanism, improve environmental economic policies;

- Improve law and regulation system, carry on supervision and management based on law;

- Establish regional coordination mechanism and integrated regional environmental management;

- Establish monitoring and warning system, cope with heavy pollution episodes;

- Clarify responsibilities of government, enterprise and society, mobilize public

participation

Detailed measures in the APPC-AP can be found in its English translation version (http://www.cleanairchina.org/product/6349.html, accessed on Mar 29, 2019) (CAAC, 2013b). To ensure that APPC-AP goals could be accomplished, China adopted a temporary measure in 2017, i.e., the intensified supervision for air pollution control in Jinjinji and around area

(http://www.gov.cn/hudong/2017-07/14/content_5210588.htm, accessed on Mar 29, 2019). There had been great achievements at the end of 2017. For examples (Zheng et al., 2018): 71% of the power plants met the ultralow emission levels; average efficiency of coal fired power units reduced from 321 gce/kWh in 2013 to 309 gce/kWh in 2017; Non-methane volatile organic compounds (NMVOC) emissions were cut down by 30% through the implementation of leak detection and

repair (LDAR) program for petrochemical industry; all coal boilers smaller than 7MW in urban areas were shut down; all "yellow label" vehicles (referring to which gasoline and diesel vehicles that fail to meet Euro 1 and Euro 3 standards, respectively) were eliminated by the end of 2017, etc.

The implementation of APPC-AP, together with 12[th] FYP on APPC-KR, had led to dramatic drop in $PM_{2.5}$ concentrations from 2013 to 2017 (Figure 8(e) and (f)). $PM_{2.5}$ trends of 2013-2017 for

entire China, Jingjinji, YRD, and PRD Regions were -4.27, -6.77, -6.36, and -2.11 μg/m$^3$/year (all $p<0.05$), respectively (Table 2). This is comparable to a recent study (Silver et al., 2018), which found that median trend in annual mean $PM_{2.5}$ concentration across all ground air pollution

monitoring stations is -3.4 μg/m$^3$/year from 2015 to 2017. Table 4 shows PM$_{2.5}$ concertation improvement goals and final accomplishments for APPC-AP. The goals required PM$_{2.5}$ concentrations in Jingjinji, YRD, and PRD Regions in 2017 should decreased by 25%, 20%, and 15% compared to 2012, and the annual mean PM$_{2.5}$ of Beijing should reach at around 60 μg/m$^3$.

Since there were no ground measurements in 2012, the Ministry of Ecology and Environment (MEE) of China used 2013 as the base year to assess the performance of APPC-AP (http://www.mee.gov.cn/gkml/sthjbgw/stbgth/201806/t20180601_442262.htm, accessed on Mar 29, 2019). To maintain consistency with the official performance assessment, we also used 2013 as the base year. Results show that the arithmetic average of satellite PM$_{2.5}$ concentrations for Jingjinji,

YRD, and PRD Regions were decreased by 36.9%, 37.1%, and 14.0%, respectively. And annual mean PM$_{2.5}$ of Beijing was 44.67 μg/m$^3$ in 2017. From the view of satellite, Jingjinji, YRD, and Beijing had accomplished their goals, and PRD was very close to the goal. However, the pollution level was still higher than WHO Air Quality IT-1 level and NAAQS (GB 3095-2012) Level 2 annual PM$_{2.5}$ standard (both 35 μg/m$^3$).

According to the official results of APPC-AP performance assessment (Table 4), PM$_{2.5}$ of Jingjinji, YRD, and PRD Regions were decreased by 39.6%, 34.3%, and 27.7%, respectively. And annual mean PM$_{2.5}$ of Beijing was 58 μg/m$^3$ in 2017. Compared to the arithmetic average satellite PM$_{2.5}$, the populations weighted average results (Table 4) are more closed to the official results. The main reason is that official performance assessment used ground measurements. However, the

spatial distribution of ground monitors is uneven. Most of the sites are distributed in populated urban areas and only a few are located in rural areas. Compared to ground monitors, satellite remote sensing has more comprehensive spatial coverage. Figure S3 shows the spatial distribution of satellite and ground PM$_{2.5}$ concentrations of 2017 in Beijing. It can be seen that the ground monitors are clustered in polluted urban centers. The cleaner north and northwest of Beijing have few sites.

Thus the population weighted results of satellite PM$_{2.5}$ are closer to the official results, but still have differences. Since satellite PM$_{2.5}$ have better spatial coverage than ground monitors, satellite PM$_{2.5}$ can better represent the spatial variation of PM$_{2.5}$ pollution. The population weighted average

satellite $PM_{2.5}$ can better represent the health impact of $PM_{2.5}$ pollution. When using ground monitors to calculate the regional mean concentrations, the weights of area and population for each site should be considered.

## 5 Discussion and Conclusions

Xue et al. (2019) developed a machine learning method to estimate $PM_{2.5}$ concentrations in China from 2000–2016. They reported that overall trends of $PM_{2.5}$ in China were 2.097 μg/m³/year ($p<0.001$), 0.299 μg/m³/year ($p>0.05$), −4.511 μg/m³/year ($p<0.001$) in 2000-2007, 2008-2013, and 2013-2016, respectively. Lin et al. (2018) estimated high-revolution $PM_{2.5}$ in annual scale in China from 2001 to 2015, and found nation-scale trends of 0.04 μg/m³/year, -0.65 μg/m³/year, -2.33 μg/m³/year in 2001-2005, 2005-2010, and 2011-2015, respectively. Overall, our satellite-based $PM_{2.5}$ trends are consistent with these two recent studies, except that we found no significant trend from 2005 to 2010 (0.41 μg/m³/year but $p>0.05$), which is different from the study of Lin et al. (2018). The main reason could be that they did not include western China in their study area. And statistical significance levels were not reported in their study, which could not allow us to know whether the trend was significant or not.

Although there have been several studies have studied the historical trends of $PM_{2.5}$ in China, few has study the relations between the trends and air pollution control policies. This paper reviewed the air pollution control policies from 2005 to 2017. And for the first time we gave an overall evaluation of the effects of these policies on $PM_{2.5}$ pollution improvement in China from the perspective of satellite remote sensing. Results show that our satellite $PM_{2.5}$ dataset is a good source to evaluate the performance of air pollution policies. The trends of satellite-derived $PM_{2.5}$ concentrations is consistent with the implementation of air pollution control policies in different periods.

The ECER policy implemented in 11[th] FYP period (see Table 1 and Section 4.3) had co-benefits on $PM_{2.5}$ pollution control. The overall $PM_{2.5}$ pollution had certain decrease (-0.56 μg/m³/year for entire China, $p=0.053$) after 2007, but the effects were limited. The Environmental

Protection Plan and ECER policy during 12$^{th}$ FYP period were basically the extension of 11$^{th}$ FYP policy, with additional total emission control on $NO_x$. However, the total emission control oriented policy had shown its limitation. The $PM_{2.5}$ concentrations of polluted areas did not decrease from 2010 to 2013 (e.g., -0.45 μg/m$^3$/year for Jingjinji Region, $p$=0.783).

To address the $PM_{2.5}$ pollution issue, China implemented two strict policies: the 12$^{th}$ FYP on APPC-KR in 2012 and APPC-AP in 2013. The goal of air quality improvement was proposed for the first time. Besides, China incorporated $PM_{2.5}$ as a major control pollutant into the National Ambient Air Quality Standard (GB 3095-2012). All these polices (details can be found in Table 1 and Sections 4.4 and 4.5) had led to dramatic decreases of $PM_{2.5}$ after 2013 (-4.27 μg/m$^3$/year for

entire China, $p$<0.001). And the implementation of these policies was also an important mark that environmental management in China began to change from total emission control oriented mode to environmental quality improvement oriented mode.

     It should be noted that inter-annual variation in meteorology has also contributed to the changes in $PM_{2.5}$. A recent study shows that meteorological conditions contributed approximately

20% of the $PM_{2.5}$ reduction in Beijing from 2013 to 2017, while the control of anthropogenic emissions contributed 80% (Chen et al., 2019). In addition, the slowdown of economic development after financial crisis in 2008 might contribute to the $PM_{2.5}$ emissions reduction. According to China Statistical Yearbook (NBS, 2018), the gross domestic products (GDP) growth rate decreased from 14.2% in 2007 to 6.9% in 2017. However, the GDP growth rates are still relatively high at current

stage (6%~7%). Contrarily, the $PM_{2.5}$ concentrations have decreased dramatically. Without effective air pollution control policies, the $PM_{2.5}$ pollution level would not decrease rapidly. Therefore, effective air pollution control policy was the main reason for $PM_{2.5}$ pollution reduction after 2013. Meteorological conditions also contributed a small portion of $PM_{2.5}$ reductions.

     The trends in $PM_{2.5}$ concentrations in China also showed spatial heterogeneity. Multiple

reasons may explain the regional differences, e.g., the pollution levels of base year, the regional differences of industrial structures, the spatial heterogeneity of anthropogenic and natural emissions, economic and industry development differences, variations of regional policies, and variations of

meteorological conditions, etc.

Currently, China has achieved great success in $PM_{2.5}$ pollution control. However, $PM_{2.5}$ concentrations in many areas are still much higher than Level 2 annual $PM_{2.5}$ standard of 35 μg/m$^3$ of GB 3095-2012, which is corresponding to WHO Air Quality IT-1 level. China has implemented a new air pollution control policy from 2018, i.e., the Three-year Action Plan to Win Battle for Blue Skies (2018-2020). China's air quality is expected to be further improved in the next three years.

This study extended the satellite $PM_{2.5}$ dataset in our previous study (Ma et al., 2016) to the year of 2017 and obtained longer time series of satellite $PM_{2.5}$ data, which can provide more spatially-resolved and high accurate $PM_{2.5}$ data for epidemiological, health impact assessment, and social economic impact studies in China.

## Authors contributions

J. B. conceived and designed the study. R. L. collected and processed the data. Z. M. and Y. L. performed statistical modeling for satellite $PM_{2.5}$ predictions. Z. M. analyzed the spatiotemporal trends of $PM_{2.5}$ concentrations. J. B. prepared and analyzed the air pollution control policies. Z. M. prepared the manuscript with contributions from all co-authors.

## Acknowledgment

This work was supported by the National Natural Science Foundation of China (91644220, 71433007, and 41601546), and the Fundamental Research Funds for the Central Universities of China (0211-14380078).

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

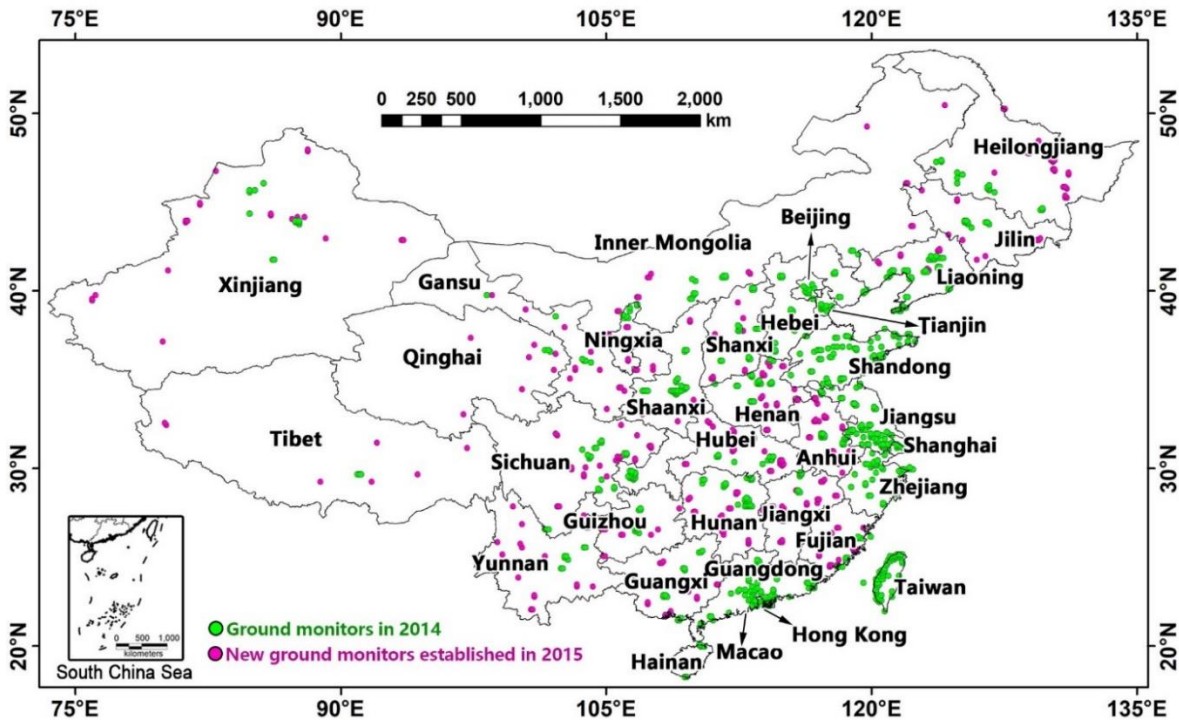

**Figure 1. Spatial distributions of ground PM₂.₅ monitors involved in model fitting and validation. Red circles denote the ground monitor in 2014. Pink circles denote new ground monitors established in 2015.**

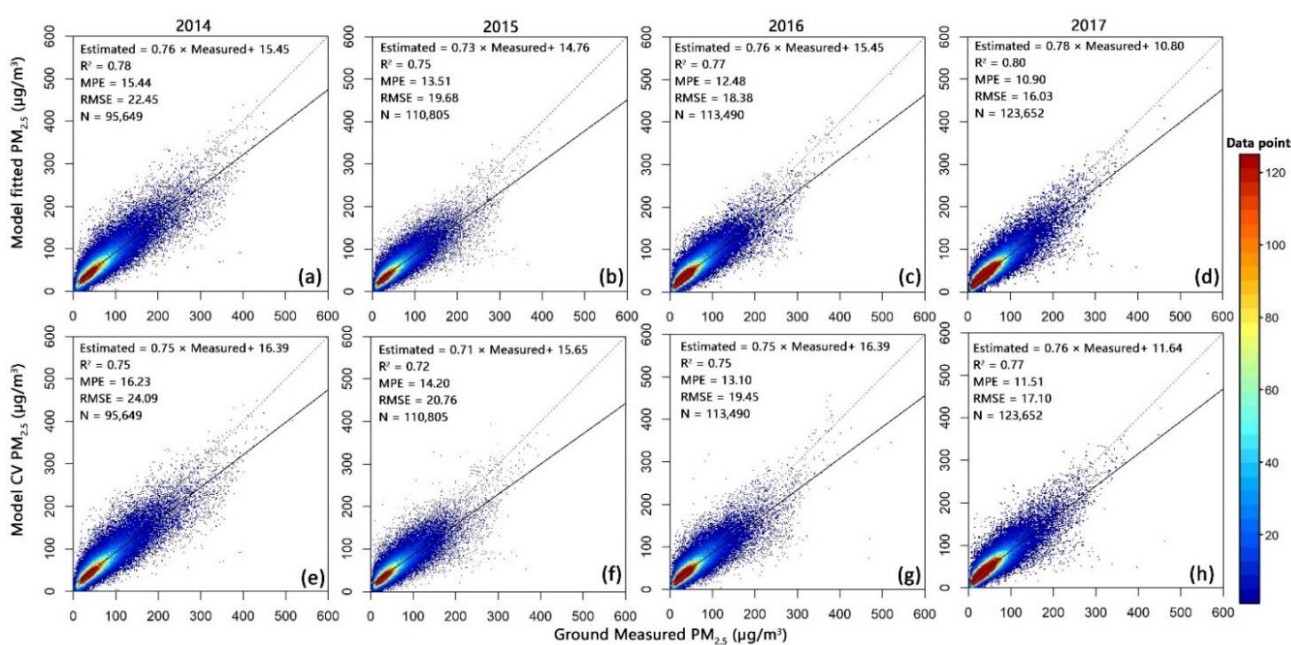

**Figure 2. Model fitting (upper row) and cross validation (CV, lower row) results for satellite PM₂.₅ prediction models from 2014 to 2017.**

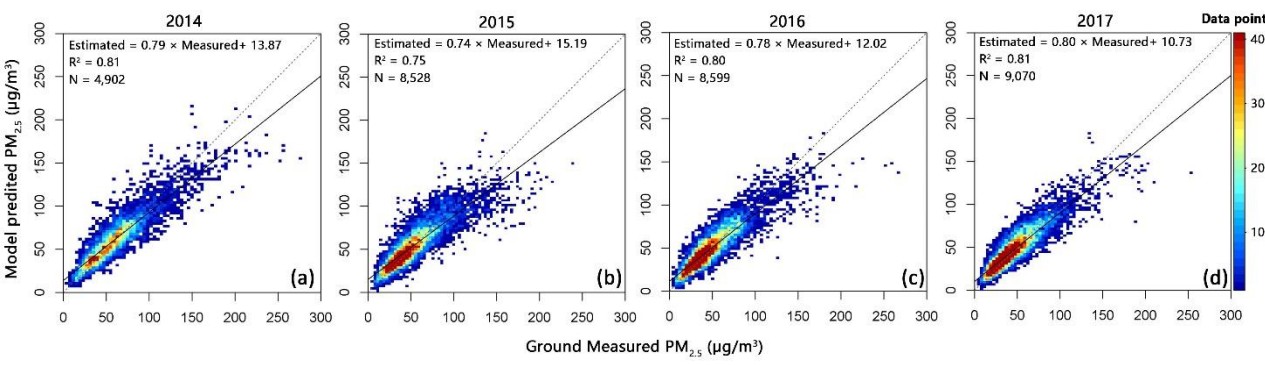

**Figure 3. Validation of monthly mean PM₂.₅ predictions from 2014 to 2017.**

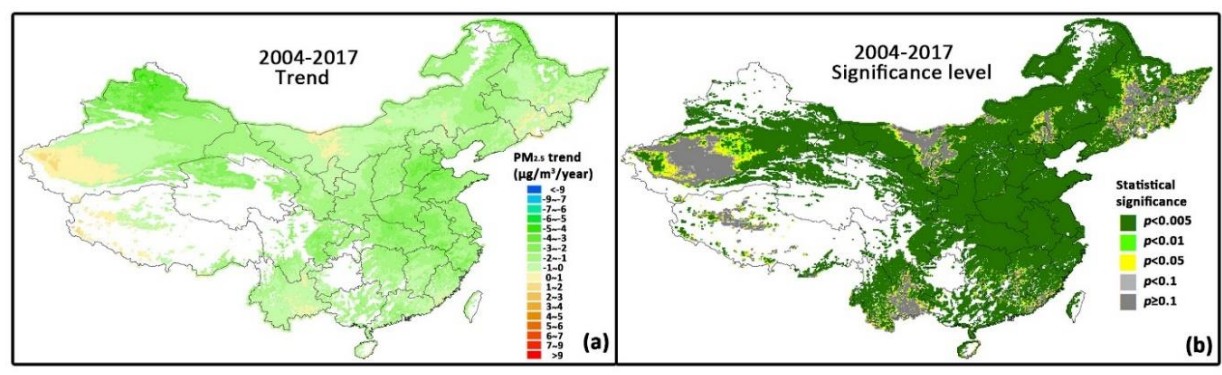

**Figure 4. Spatial distributions of annual mean satellite-derived PM$_{2.5}$ concentrations from 2004 to 2017.**

**Figure 5. Spatial distributions of PM$_{2.5}$ trends and significance levels in China from 2004 to 2017.**

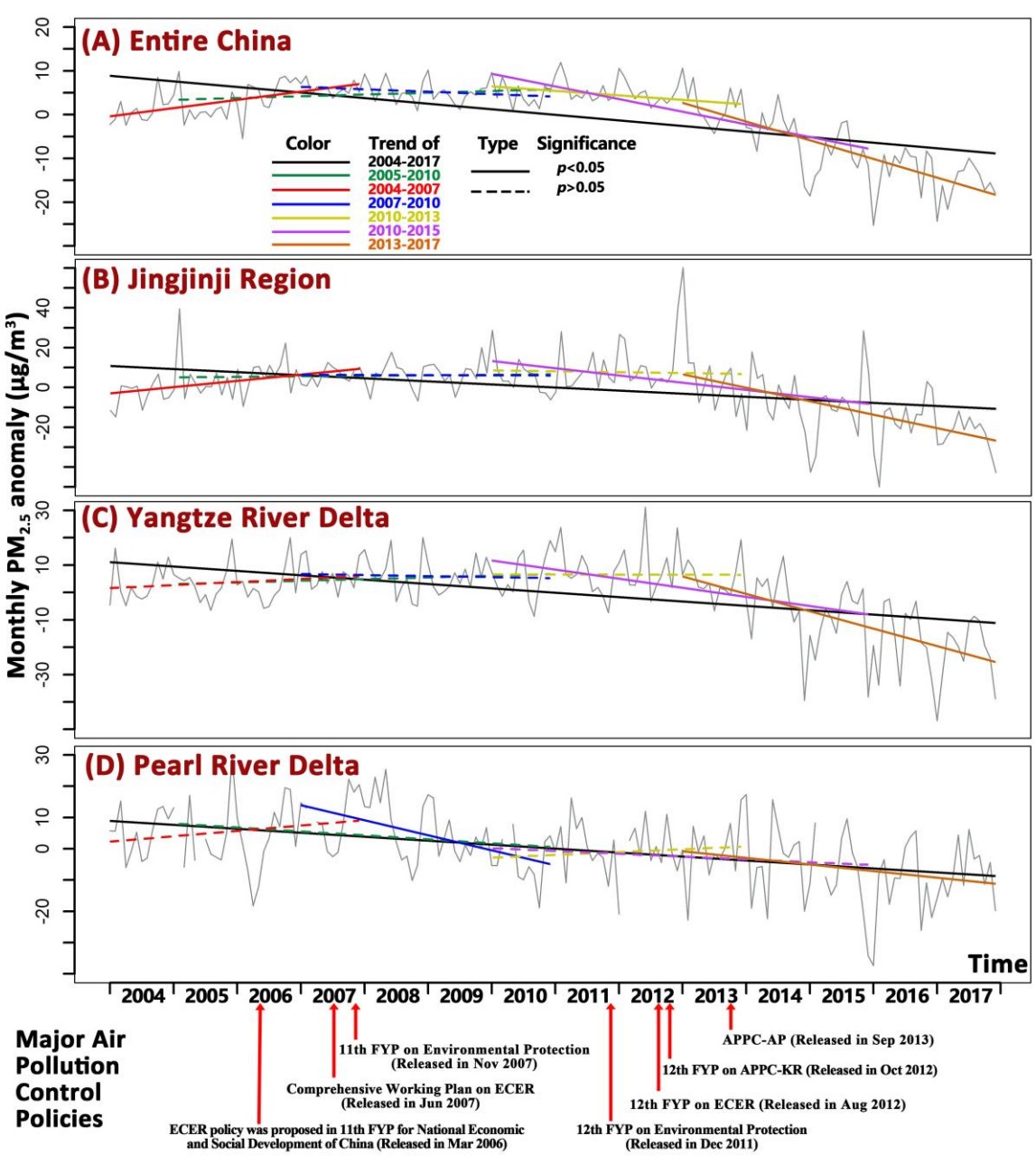

**Figure 6. PM₂.₅ trends of entire China, Jingjinji, Yangtze River Delta (YRD), and Pearl River Delta (PRD) Regions from 2004 to 2017, and corresponding air pollution control policies**

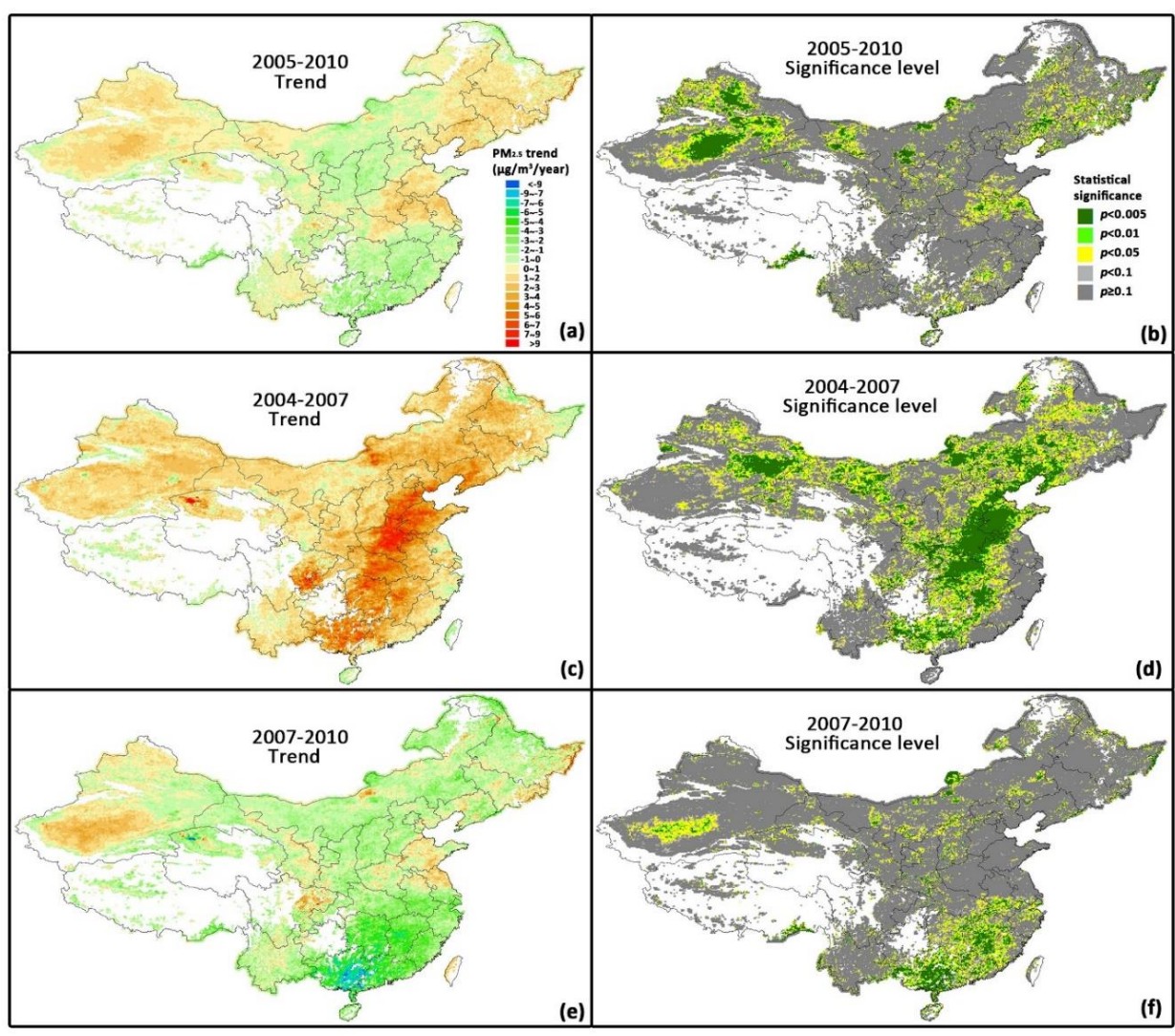

**Figure 7. Spatial distributions of PM₂.₅ trends and significance levels in China from 2005 to 2010.**

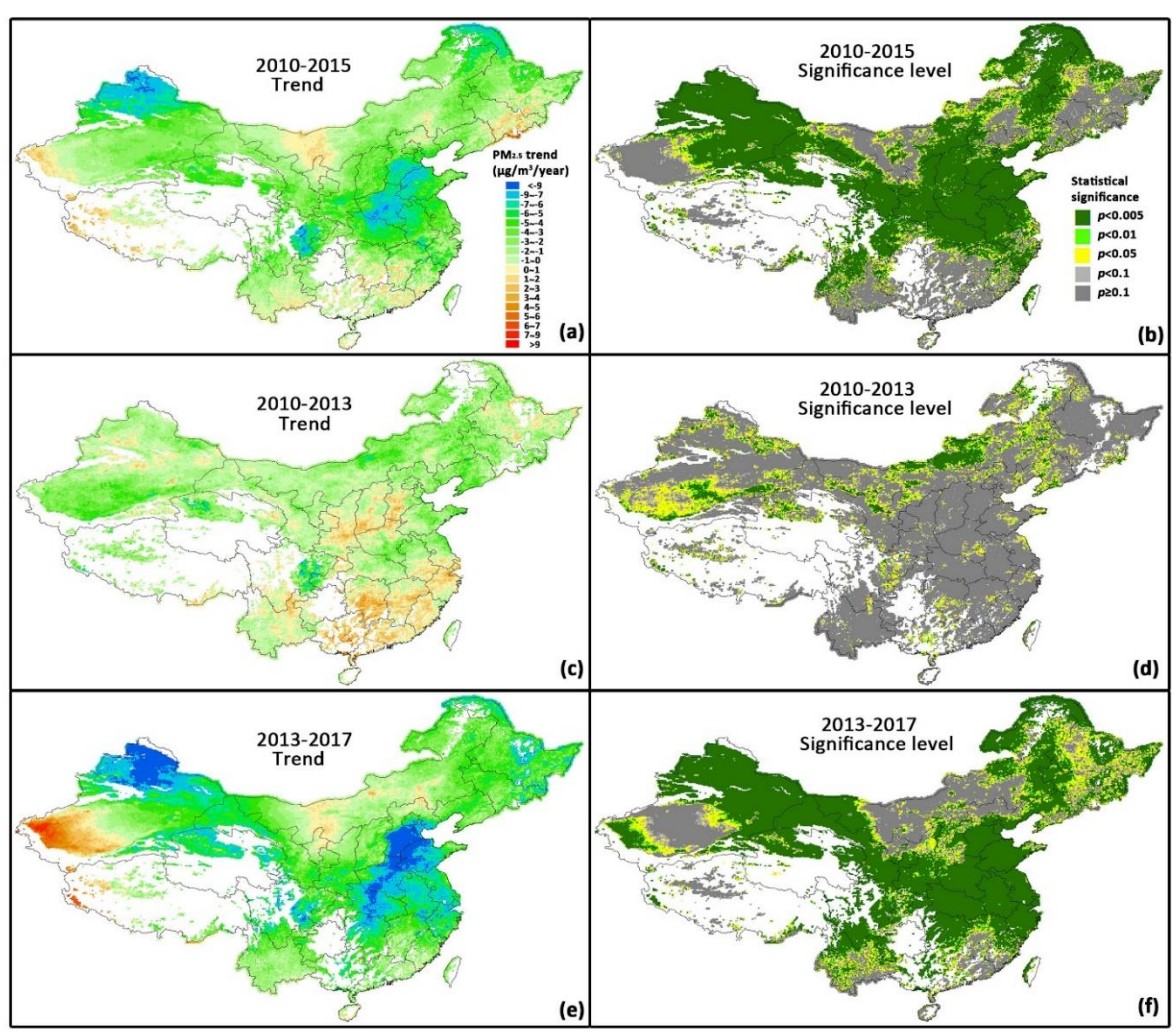

**Figure 8. Spatial distributions of PM₂.₅ trends and significance levels in China from 2010 to 2017.**

**Table 1. Overview of major air pollution control policies in China from 2005 to 2017**

| Policy [a] | Base year | Implementation period | Major goals (Compared to base year) | Major measures |
|---|---|---|---|---|
| 11th FYP on Environmental Protection | 2005 | 2006-2010 | • $SO_2$ emission should reduce by 10% | • Implement of desulphurization projects of coal-fired power plants<br>• Prevent and control urban $PM_{10}$ pollution, relocate pollution industrial plants in urban areas, control construction and road dust<br>• Implement total emission control policy for key industrial pollution sources, control emission of sulfur dioxide and soot (dust)<br>• Strengthen vehicle pollution prevention and control, improve quality and efficiency of gasoline |
| ECER during 11th FYP | 2005 | 2006-2010 | • Energy consumption per GDP capita should decrease by20%<br>• $SO_2$ emission should reduce by 10% | • Promote industrial and energy structure adjustment, restrain the development of industries with high energy consumption and pollution, eliminate backward production capacity, promote production capacity with low energy consumption and low pollution<br>• Implement ten major energy conservation projects, implement desulphurization projects of coal-fired power plants |
| 12th FYP on Environmental Protection | 2010 | 2011-2015 | • $SO_2$ emission should reduce by 8%<br>• $NO_x$ emission should reduce by 10% | • Implement desulphurization and denitration facilities for coal-fired power sector and major industrial sectors<br>• Control NOx emissions of vehicles and ships<br>• Deepen PM and VOCs pollution control<br>• Promote urban air pollution prevention and control, implement coordinated control of various pollutants in key areas, monitor $PM_{2.5}$ and $O_3$ in Jingjinji, Yangtze River Delta, and Pearl River Delta regions |
| ECER during 12th FYP | 2010 | 2011-2015 | • Energy consumption per GDP capita should decrease by16%<br>• $SO_2$ emission should reduce by 8%<br>• $NO_x$ emission should reduce by 10% | • Adjust and optimize industrial structure, control the development of industries with high energy consumption and pollution, eliminate backward production capacity<br>• Adjust energy consumption structure, strengthen energy conservation for industrial, building, transportation, commercial and civil areas, etc.<br>• Strengthen emissions reduction in key industrial sectors, promote desulphurization and denitration, control emissions of vehicles, promote the control of $PM_{2.5}$ |
| The 12th FYP on APPC-KR [b] | 2010 | 2011-2015 | • Emission of the $SO_2$, NOx, and industrial PM should decrease 12%, 13%, and 10%, respectively<br>• The annual average concentration of $PM_{10}$, $SO_2$, $NO_2$ and $PM_{2.5}$ should decrease by 10%, 10%, 7%, and 5%, respectively | • Identify the key regions and implement regional specific management<br>• Strictly control high energy consumption and high pollution projects, control new pollutants emissions, implement strict emission standard, and enhance control requirements of VOCs in key regions<br>• Strengthen elimination of backward production capacity, optimize industrial layout<br>• Optimize energy consumption structure, develop clean energy, control total coal consumption, establish restricted zones for high polluting fuels, eliminate small coal boilers, promote clean and efficient utilization of coal<br>• Comprehensively implement co-control of multiple pollutants ($SO_2$, $NO_x$, PM, VOCs), strengthen vehicle pollution prevention and control<br>• Innovate regional management mechanism, establish joint regional prevention and control coordination mechanism, establish and perfect ground monitoring network |
| APPC-AP | 2012 | 2013-2017 | • $PM_{2.5}$ concentrations of Jingjinji, Yangtze River Delta, and Pearl River Delta regions should reduce by 25%, 20%, and 15% respectively<br>• $PM_{2.5}$ concentrations of Beijing should be controlled at around 60 μg/m³ | • Enhance comprehensive air pollution control on industrial enterprises, deepen non-point source control, strengthen vehicle pollution control<br>• Adjust, optimize, and upgrade industrial structure, strictly control new capacity with high energy consumption and high pollution, accelerate elimination of backward production capacity, reduce excess capacity<br>• Accelerate energy structure adjustment, accelerate utilization of clean energy, control total coal consumption, promote clean utilization of coal, improve energy efficiency<br>• Optimize industrial layout<br>• Utilize the market mechanism, improve the pricing and tax policy, establish regional coordination mechanism<br>• Establish monitoring, early warning, and emergency system for heavy pollution episodes |

[a] Abbreviations: FYP: Five Year Plan; ECER: Energy Conservation and Emissions Reduction; APPC-KR: Air Pollution Prevention and Control in Key Regions; APPC-AP: Action Plan of Air Pollution Prevention and Control

[b] The key regions are shown in Figure S1 (Supplemental Materials)

**Table 2. Trends and 95% confidence intervals (CI) of PM$_{2.5}$ concentrations for entire China, Jingjinji, Yangtze River Delta, and Pearl River Delta Regions from 2004 to 2017**

| Period | Trend | Entire China | Jingjinji Region | Yangtze River Delta | Pearl River Delta |
|---|---|---|---|---|---|
| 2004-2017 | Trend (μg/m$^3$/year) | **-1.27** | **-1.55** | **-1.60** | **-1.27** |
| | 95% CI (μg/m$^3$/year) | **(-1.50, -1.04)** | **(-2.06, -1.03)** | **(-2.02, -1.18)** | **(-1.66, -0.88)** |
| | Significance | ***p<0.001*** | ***p<0.001*** | ***p<0.001*** | ***p<0.001*** |
| 2005-2010 | Trend (μg/m$^3$/year) | 0.41 | 0.26 | 0.61 | -1.26 |
| | 95% CI (μg/m$^3$/year) | (-0.01, 0.82) | (-0.83, 1.36) | (-0.31, 1.54) | (-2.73, 0.21) |
| | Significance | $p$=0.055 | $p$=0.633 | $p$=0.191 | $p$=0.091 |
| 2004-2007 | Trend (μg/m$^3$/year) | **1.88** | **3.14** | 1.12 | 1.72 |
| | 95% CI (μg/m$^3$/year) | **(1.12, 2.64)** | **(1.07, 5.22)** | (-0.51, 2.74) | (-0.79, 4.23) |
| | Significance | ***p<0.001*** | ***p<0.005*** | $p$=0.174 | $p$=0.174 |
| 2007-2010 | Trend (μg/m$^3$/year) | -0.56 | -0.08 | -0.37 | **-4.81** |
| | 95% CI (μg/m$^3$/year) | (-1.12, 0.01) | (-1.80, 1.64) | (-2.10, 1.35) | **(-7.06, -2.55)** |
| | Significance | $p$=0.053 | $p$=0.927 | $p$=0.664 | ***p<0.001*** |
| 2010-2013 | Trend (μg/m$^3$/year) | **-1.03** | -0.45 | -0.04 | 0.89 |
| | 95% CI (μg/m$^3$/year) | **(-1.84, -0.21)** | (-3.73, 2.83) | (-2.16, 2.08) | (-1.34, 3.13) |
| | Significance | ***p<0.050*** | $p$=0.783 | $p$=0.970 | $p$=0.425 |
| 2010-2015 | Trend (μg/m$^3$/year) | **-2.89** | **-3.63** | **-3.33** | -0.90 |
| | 95% CI (μg/m$^3$/year) | **(-3.50, -2.28)** | **(-5.59, -1.68)** | **(-4.76, -1.89)** | (-2.34, 0.54) |
| | Significance | ***p<0.001*** | ***p<0.001*** | ***p<0.001*** | $p$=0.219 |
| 2013-2017 | Trend (μg/m$^3$/year) | **-4.27** | **-6.77** | **-6.36** | **-2.11** |
| | 95% CI (μg/m$^3$/year) | **(-5.20, -3.34)** | **(-9.46, -4.07)** | **(-8.38, -4.34)** | **(-4.14, -0.09)** |
| | Significance | ***p<0.001*** | ***p<0.001*** | ***p<0.001*** | ***p<0.050*** |

**Table 3. Goals accomplishments for key regions of 12th FYP on APPC-KR**

| Region | Goal (Decreased by) | Average satellite PM$_{2.5}$ concentrations | | | Population weighted average satellite PM$_{2.5}$ concentrations | | |
|---|---|---|---|---|---|---|---|
| | | 2010 (μg/m$^3$) | 2015 (μg/m$^3$) | Decreased by | 2010 (μg/m$^3$) | 2015 (μg/m$^3$) | Decreased by |
| Beijing | 15% | 68.75 | 58.47 | 14.9% | 83.41 | 70.61 | 15.3% |
| Tianjin | 6% | 97.17 | 75.17 | 22.6% | 96.13 | 76.09 | 20.8% |
| Hebei | 6% | 74.72 | 58.19 | 22.1% | 101.25 | 75.15 | 25.8% |
| Shanghai | 6% | 66.41 | 58.83 | 11.4% | 64.30 | 60.67 | 5.7% |
| Jiangsu | 7% | 81.23 | 62.24 | 23.4% | 82.18 | 63.19 | 23.1% |
| Zhejiang | 5% | 52.85 | 38.73 | 26.7% | 58.68 | 47.37 | 19.3% |
| Pearl River Delta | 5% | 45.00 | 37.97 | 15.6% | 50.07 | 40.99 | 18.1% |
| Central Liaoning | 6% | 58.10 | 53.00 | 8.8% | 64.97 | 58.40 | 10.1% |
| Shandong | 7% | 94.57 | 71.83 | 24.0% | 97.83 | 73.76 | 24.6% |
| Wuhan Region | 5% | 75.02 | 55.41 | 26.1% | 79.86 | 58.62 | 26.6% |
| Changzhutan Region | 5% | 64.81 | 52.75 | 18.6% | 72.32 | 60.19 | 16.8% |
| Chongqing | 6% | 65.89 | 47.48 | 27.9% | 77.36 | 52.71 | 31.9% |
| Chengdu Region | 5% | 83.55 | 52.22 | 37.5% | 92.22 | 57.40 | 37.8% |
| Fujian | 4% | 37.42 | 28.02 | 25.1% | 34.48 | 29.22 | 15.3% |
| Central and Northern Shanxi | 4% | 53.76 | 40.05 | 25.5% | 63.05 | 46.78 | 25.8% |
| Guanzhong | 4% | 65.91 | 45.33 | 31.2% | 79.54 | 53.91 | 32.2% |
| Lanzhou Region | 4% | 55.42 | 45.31 | 18.2% | 62.47 | 47.77 | 23.5% |
| Yinchuan | 5% | 42.81 | 48.14 | -12.4% | 46.51 | 51.81 | -11.4% |
| Urumqi Region | 4% | 60.26 | 27.83 | 53.8% | 65.80 | 36.05 | 45.2% |

**Table 4. Goal accomplishments of APPC-AP (2013-2017)**

| Region | Goal (Decreased by) | Official assessment results [a] | Average satellite PM$_{2.5}$ concentrations | | | Population weighted average satellite PM$_{2.5}$ concentrations | | |
|---|---|---|---|---|---|---|---|---|
| | | | 2013 (μg/m$^3$) | 2017 (μg/m$^3$) | Decreased by | 2013 (μg/m$^3$) | 2017 (μg/m$^3$) | Decreased by |
| Jingjinji | 25% | 39.6% | 76.01 | 47.98 | 36.9% | 100.91 | 60.97 | 39.6% |
| Yangtze River Delta | 20% | 34.3% | 66.60 | 41.87 | 37.1% | 71.98 | 46.45 | 35.5% |
| Pearl River Delta | 15% | 27.7% | 45.15 | 38.84 | 14.0% | 49.96 | 40.37 | 19.2% |
| Beijing | Be controlled at around 60 μg/m$^3$ | 58 μg/m$^3$ | 68.20 | 44.67 | 34.5% | 82.69 | 55.07 | 33.4% |

[a] See http://www.mee.gov.cn/gkml/sthjbgw/stbgth/201806/t20180601_442262.htm, accessed on Mar 29, 2019