# Peer review of "Effects of air pollution control policies on $PM_{2.5}$ pollution improvement in China from 2005 to 2017: a satellite based perspective"

_Atmospheric Chemistry and Physics, 2018_

## Referee Comment (RC1) · Anonymous Referee #1 · 11 Dec 2018

This is an interesting paper putting forward a historical perspective on PM2.5 surface concentrations in China. The authors propose a statistical method that relates satellite-observed aerosol optical depth (AOD) over China to measurements of PM2.5 at the surface. The authors use the years 2013 and later when both satellite and surface measurements were available to train their method. Prior to 2013 there was no ground-based network to speak of in China.

Then the method, essentially a multivariate regression of information on the atmospheric state, is applied to infer surface PM2.5 for the entire 2005-2017 period from MODIS AOD This allows the authors to evaluate the effectiveness of the various Chi-

nese air pollution control policies that have been applied in this period. Based on the satellite-estimated PM2.5 trends, the authors claim a "periodic victory" for Chinese policies to clean up the air.

I think the paper addresses a relevant topic that is appropriate for publication in ACP, but I have concerns about the method, which is not well described in this manuscript. Also for non-Chinese readers, it would be necessary to clarify what the various Chinese policies consisted of. We read very little about what measures were actually implemented, and how they may have had an effect. This is important information to share with an eye on other countries going through a rapid development phase, and wishing to limit the effects of air pollution. The authors owe it to the world, so to speak.

**Specific comments**

The abstract is not very clear. There are many abbreviations referring to policies applicable in certain periods only that will not be immediately clear or well-known to the wide readership of ACP. The authors should rewrite their abstract with a focus on storytelling how Chinese PM2.5 changes over time between 2005 and 2017, and why. The results summarized should be presented in quantitative fashion.

**Methods**

The method to infer PM2.5 from MODIS AOD is explained only very briefly with repeated reference to a previous paper by the same authors. For this paper to standalone, the authors should provide much more detail on how their statistical "two-stage" method works, and how robust the method is. The authors should briefly explain what drives the relationship between PM2.5 and AOD. Which parameters explain most of he variance and why.

Specifically:

1. Provide the equation establishing the relationship between AOD and surface PM2.5

2. Explain how the fit parameters have been derived, and discuss the orthogonality of

the various explanatory terms (humidity, boundary layer height, T, . . .)

3. Discuss the temporal resolution of the relationship ("each year's model")

4. Discuss the differences and agreement with the model-scaling approach

Since the method relies on the quality of the MODIS and surface PM2.5 data, these aspects should be discussed as well.

Related to the lack of information on the method, are the terms "random intercepts" and "random slopes" mentioned on page 7. Without reading the previous paper by the authors in a different journal, it is entirely unclear what these terms mean. It shows that this manuscript cannot be read on its own, which is not the standard for a paper in ACP.

Related to the trends, it is unclear how the trends were determined. Did they use a linear model of the form $y = a + b\,t$, how did they deal with seasonality, weighing of sparsely sampled months, etc.? They need to provide more detail and also include figures showing the temporal evolution of the PM2.5 estimates, along with the satellite data, and ground-based observations for one or a few particular locations.

Section 2 on the policies is too technocratic. We read about the official titles of the policies, but the authors should make clear not just in (the valuable but too long) Table 1 but also in the main text what the policies consisted of. I realize they cannot be exhaustive all the time, but they should provide an assessment of what they think were the most effective measures taken under a certain policy, and the evidence to back this up. This is important to make a convincing case, and allows others to learn from the policies taken. One suggestion is to come up with a figure showing a timeline of the various measures and their anticipated effect on Chinese PM2.5 levels. Such a figure could then later be confronted with the observed PM2.5 evolution, and tell the story whether measures have been effective.

**Minor comments**

[Figure]

P2, L3: polices → policies

P3, L13-14: the citations are quite China-centric. Consider citing studies on SO2 and NO2 trends over China from non-Chinese groups, e.g. Itahashi et al. [2012], Krotkov et al. [2016], Miyazaki et al. [2017].

P4, L11: pollution(s)

P4, L26: policy, not policies

P5, L21: unclear what $R^2 refers to.$

P5, L25: "Validation results indicated..." be more specific. Validation done where, when?

P6, L12: suggest to remove referring to Ma et al. [2016]. This paper should describe the method briefly itself.

P6, L20-21: please discuss the representativeness of the PM2.5 stations for the size of a MODIS pixel, or vice versa?

P9, L25: grammar

P10, L9-10: "strengthened the ECER measures" → explain how

P10, L11-12: explain qualitatively how this would have worked

P10, L18-19: explain why further reduction emissions had no beneficial effect anymore

P10, L21-22: rephrase .. I don't think bottleneck is the term you should use.

P10, L25: "After that" ... → be more specific what the policy consisted of then

P11, section 4.4: it would be useful to include here already how the findings relate to Chinese and WHO air quality standards.

P11, L6-7: how? we remain in te dark what was actully done and how that helped

P11, L11: what explains the regional differences?

P11, L21: close(d)

P11, L25: what are the "official results"?

P12, L20: "the overall decrease" → be quantitative

P13, L2: "All these policies" → it should be made clear what was the essence of this

P13, L4: MEE → ?

P13, L9: "air pollution control in China has achieved a periodic victory" → this is awkward, do the authors mean that the measures taken so far have resulted in a temporary solution, or, more precisely, have succeeded to mitigate the worst aspects of PM2.5 pollution?

Figure 2: unclear what difference is between upper and lower rows.

**Suggested refereces**

Krotkov, N. A., McLinden, C. A., Li, C., Lamsal, L. N., Celarier, E. A., Marchenko, S. V., Swartz, W. H., Bucsela, E. J., Joiner, J., Duncan, B. N., Boersma, K. F., Veefkind, J. P., Levelt, P. F., Fioletov, V. E., Dickerson, R. R., He, H., Lu, Z., and Streets, D. G.: Aura OMI observations of regional SO2 and NO2 pollution changes from 2005 to 2015, Atmos. Chem. Phys., 16, 4605-4629, doi:10.5194/acp-16-4605-2016, 2016.

Miyazaki, K., Eskes, H., Sudo, K., Boersma, K. F., Bowman, K., and Kanaya, Y.: Decadal changes in global surface NOx emissions from multi-constituent satellite data assimilation, Atmos. Chem. Phys., 17, 807-837, https://doi.org/10.5194/acp-17-807-2017, 2017.

Itahashi, S., Uno, I., Yumimoto, K., Irie, H., Osada, K., Ogata, K., Fukushima, H., Wang, Z., and Ohara, T.: Interannual variation in the fine-mode MODIS aerosol optical depth and its relationship to the changes in sulfur dioxide emissions in China between 2000

and 2010, Atmos. Chem. Phys., 12, 2631-2640, https://doi.org/10.5194/acp-12-2631-2012, 2012.

---

## Referee Comment (RC2) · Anonymous Referee #2 · 26 Feb 2019

General comments The paper provides a useful overview of recent air quality control policies in China, while using an independent source of data to assess their efficacy. A statistical method is used to correlate satellite retrievals of Aerosol Optical Depth (AOD) to ground level PM2.5 in China, by correlating AOD with meteorological data, fire spots and forest cover. It uses the large network of Chinese measurement stations to verify the model. The 2013 model, which was developed in another paper (Ma et al 2016) is used to project the concentration of PM2.5 backwards to 2005, while a separate model is developed each year for 2014 - 2017. This gives a 13-year PM2.5 dataset with complete spatial and temporal coverage for 2005 – 2017, which is then used to assess the success of China's air quality control policy that underwent significant changes during

this period. Linear trends are calculated for the periods corresponding to specific policies (e.g. Five Year Plans). Calculated PM2.5 concentrations are also compared with official government data, to verify that targets were met. While this retrospective analysis of the success of China's control of PM2.5 pollution is very useful, the authors need to ensure that they acknowledge the role that inter-annual variation in meteorology may play in these trends.

Specific Comments Abstract The majority of the abstract summarises the discussion section. A brief description of the two stage statistical model, including its predictors could be added. Intro P3, L23: It may be worth adding a sentence that briefly explains what the 'scaling method' is. There is a citation to Liu 2014 to back up the statement that, "Compared to the scaling method, statistical models have greater prediction accuracy but require large amount ground-measured PM2.5 data to develop the statistical models (Liu, 2014)". However, there is not a reference that corresponds to the "Liu, 2014" citation. Since the justification of method choice relies on this reference, it should be added before the paper is reviewed again. Overview of air pollution control policies in China from 2005 to 2017 This section is a very broad summary of the actions within Five Year Plans and other major government directives that are relevant to air pollution control. The specific policies (e.g. 'Implement desulphurization and denitration facilities for coal-fired power sector and major industrial sectors') are summarised in Table 1, along with the metrics by which the policies' success will be judged. It may be useful to, where possible, cite government press releases/reports or literature that assess the success of these policies. However, the text in this section does not make any mention of the policies listen in Table 1. It would be useful for the reader for some information from Table 1 to be synthesised into this section, along with citations to previous studies that have attempted to assess the success of these policies (e.g. Schreifels et al, 2012) P5, L13. It may be worth defining what China's 'new air quality standard' here, where it is first mentioned. It may be useful to provide the old air quality standard, and the name of the standard (GB 3095-2012). Currently the actual threshold number of China's air quality standard is first referenced of P13, L10 in

the conclusion. Data and Method P6, L19: Paper uses PM2.5 data from the CNEMC. Other papers, (e.g. Rohde and Muller (2015); Liu et al (2016)) have noted quality issues with this data. Were any quality control procedures applied to this data? Since the ground monitoring stations are typically within urban areas, could this bias the statistical model so that the PM2.5 predictions for non-urban areas is inaccurate? Why use the updated data to create separate statistical models for 2014, 2015, 2016 and 2017, yet only use the 2013 model to project back the PM2.5? Why should the 2013 model be more appropriate than the other years? Why not combine all the years where measurements are available? How is it justified to fit the model separately to the data in each province? Isn't using province boundaries somewhat arbitrary? Many other studies of trends in atmospheric concentrations use a non-parametric trend estimator such as the Thiel-Sen slope estimator. The authors should justify their choice of the least squares regression to estimate the slope of the trend. In the results section, and Figures 6 & 7, a p threshold of 0.1 is mentioned, but you do not mention in the methods which statistical test you used to check the significance of your trends. Some of these questions about the methodology can be answered by reading the author's previous Ma et al 2016 paper, which is published in Environmental Health Perspectives. I recommend the authors reduce their reliance on referring to this previous paper, so that the methods section in the current paper can be understood without referring to another paper which the reader will not necessarily have access to. P5, L26: Is it useful to the reader to list 9 studies that have referenced your previous paper? This list includes studies that this paper's co-authors are also co-authors on. Results and Discussions Is it really useful to compare the PM2.5 trend with the corresponding FYP policies? This suggests that policies have immediate effects, and that they are the main contributor to the trends in PM2.5. There are other important confounding factors such as inter-annual variation in meteorology, China's economic output etc. May be best to avoid statements on the effectiveness of certain policies, or mention the above caveats in the conclusion. I suggest the authors add a comparison of their results with other research that quantifies the trend in PM2.5 derived AOD in China, such as Lin et al., 2017. It

may be interesting to perform a non-linear trend analysis on this dataset in certain key regions (e.g. Jing-Jin-Ji or PRD). As you break down the trend into multiple overlapping periods of different lengths, it is difficult to get an overall impression of the rises and falls in the trend in different regions. Alternatively, a figure could be added with the yearly or monthly deseasonalised PM2.5 (averaged by different regions). I suggest the authors also mention the possibility of contribution of natural sources of aerosol to the trends. At P10, L16, the authors mention that the majority of the trend in PM2.5 during 2010-2013 are driven by decreases in Xinjiang and Central Inner Mongolia, which are both desert regions where the PM2.5 likely has a high dust component. This can be seen in your results. For example in panel (e) of Figure 7, where the western half of the Taklamakan desert has a strong positive trend, despite it being unlikely that there are large changes in emissions in this unpopulated region. Minor comments P3, L8: "However, the Chinese government did not realize the PM2.5 issues until 2012." This sentence seems disingenuous and qualitative so should be removed or rephrased. P4, L6: Remove or replace the word 'preliminary' P5 L14. "These policies indicated that the air pollution control in China began to focus on air quality improvement." This sentence could be rephrased, as it is currently seems tautological. P10, L22: The sentence "As the further development of social economic, the ECER policy had shown its bottleneck for PM2.5 reductions." does not make sense. Bottleneck may be the wrong word to describe this. P12, L25. Change 'to addressed' to "to address." P13, L6. 'Temporal' is not the right word here. Should be temporary? References Lin, C. Q., Liu, G., Lau, A. K. H., Li, Y., Li, C. C., Fung, J. C. H., & Lao, X. Q. (2018). High-resolution satellite remote sensing of provincial PM2. 5 trends in China from 2001 to 2015. Atmospheric Environment, 180, 110-116. Liu, Jianzheng, Weifeng Li, and Jie Li. "Quality screening for air quality monitoring data in China." Environmental pollution216 (2016): 720-723. Rohde, Robert A., and Richard A. Muller. "Air pollution in China: mapping of concentrations and sources." PloS one10.8 (2015): e0135749. Schreifels, Jeremy J., Yale Fu, and Elizabeth J. Wilson. "Sulfur dioxide control in China: policy evolution during the 10th and 11th Five-year Plans and lessons for the future." Energy Policy48 (2012):

779-789.

---

## Author Comment (AC1) · 31 Mar 2019

Responses to RC1

Please find the supplement for the revised manuscript and supplementary materials. We have highlighted the revisions in red font in the revised manuscript.

Comments from RC1: This is an interesting paper putting forward a historical perspective on PM2.5 surface concentrations in China. The authors propose a statistical method that relates satellite observed aerosol optical depth (AOD) over China to measurements of PM2.5 at the surface. The authors use the years 2013 and later when

both satellite and surface measurements were available to train their method. Prior to 2013 there was no ground based network to speak of in China. Then the method, essentially a multivariate regression of information on the atmospheric state, is applied to infer surface PM2.5 for the entire 2005-2017 period from MODIS AOD. This allows the authors to evaluate the effectiveness of the various Chinese air pollution control policies that have been applied in this period. Based on the satellite-estimated PM2.5 trends, the authors claim a "periodic victory" for Chinese policies to clean up the air. I think the paper addresses a relevant topic that is appropriate for publication in ACP, but I have concerns about the method, which is not well described in this manuscript. Also for non-Chinese readers, it would be necessary to clarify what the various Chinese policies consisted of. We read very little about what measures were actually implemented, and how they may have had an effect. This is important information to share with an eye on other countries going through a rapid development phase, and wishing to limit the effects of air pollution. The authors owe it to the world, so to speak.

Response: We would like to thank the reviewer for his valuable comments. We have added descriptions of the method. We also incorporated descriptions of major air pollution control measures in the main text. Please see the following responses for details.

(1) The abstract is not very clear. There are many abbreviations referring to policies applicable in certain periods only that will not be immediately clear or well-known to the wide readership of ACP. The authors should rewrite their abstract with a focus on storytelling how Chinese PM2.5 changes over time between 2005 and 2017, and why. The results summarized should be presented in quantitative fashion.

Response: The abstract has been revised according to this comment and the comment from another referee. We simply explained how these policies can impact the PM2.5 pollution. The trends analysis was revised and presented in a quantitative way. A brief description of the model has been added. Please see Abstract in P2.

(2) The method to infer PM2.5 from MODIS AOD is explained only very briefly with

repeated reference to a previous paper by the same authors. For this paper to stan-dalone, the authors should provide much more detail on how their statistical "two-stage" method works, and how robust the method is. The authors should briefly explain what drives the relationship between PM2.5 and AOD. Which parameters explain most of the variance and why. Specifically: Provide the equation establishing the relationship between AOD and surface PM2.5 Explain how the fit parameters have been derived, and discuss the orthogonality of the various explanatory terms (humidity, boundary layer height, T, : : :) Discuss the temporal resolution of the relationship ("each year's model") Discuss the differences and agreement with the model-scaling approach Since the method relies on the quality of the MODIS and surface PM2.5 data, these aspects should be discussed as well.

Response: According to the comments, we have made corresponding revisions as fol-lows: 1) We added details about the equations of the two-stage model, please see P8-P9; 2) In original manuscript, we have discussed the model performance for each year, see Lines 9-21, P10 in Section 4.1. In revised manuscript, we added the provin-cial fixed effects, model fitting, and CV results of the first-stage LME model for each year in Tables S2-S5 (Supplementary Materials). And we have discussed it in Line 22, P10 Line 5, P11. 3) A brief description of scaling method was added (Line 25, P3 Line 1, P4). We compared the model performance with previous scaling method studies, see Lines 11-17, P11. 4) Lines 23-25, P7 shows the quality of MODIS AOD data. Lines 17-21, P7 added the issues of PM2.5 data quality.

(3) Related to the lack of information on the method, are the terms "random intercepts" and "random slopes" mentioned on page 7. Without reading the previous paper by the authors in a different journal, it is entirely unclear what these terms mean. It shows that this manuscript cannot be read on its own, which is not the standard for a paper in ACP.

Response: We added details about the equations of the two-stage model, please see P8-P9.

(4) Related to the trends, it is unclear how the trends were determined. Did they use a linear model of the form y = a + b t, how did they deal with seasonality, weighing of sparsely sampled months, etc.? They need to provide more detail and also include figures showing the temporal evolution of the PM2.5 estimates, along with the satellite data, and ground-based observations for one or a few particular locations.

Response: We added details about the method for trend analysis in Lines 14-24, P9. For seasonality, we have described how we dealt with it in our original manuscript. See Lines 10-12, P9. We deseasonalized the monthly PM2.5 time series by calculating the monthly PM2.5 anomaly time series for each grid cell to remove the seasonal effect.

(5) Section 2 on the policies is too technocratic. We read about the official titles of the policies, but the authors should make clear not just in (the valuable but too long) Table 1 but also in the main text what the policies consisted of. I realize they cannot be exhaustive all the time, but they should provide an assessment of what they think were the most effective measures taken under a certain policy, and the evidence to back this up. This is important to make a convincing case, and allows others to learn from the policies taken. One suggestion is to come up with a figure showing a timeline of the various measures and their anticipated effect on Chinese PM2.5 levels. Such a figure could then later be confronted with the observed PM2.5 evolution, and tell the story whether measures have been effective.

Response: Revisions have been made according to the comments. First, we described major air pollution control measures, corresponding achievements, and how these policies were associated with PM2.5 pollutions in the main text. Such as Lines 4-13, P13; Lines 22-27, P13; Lines 14-21, P14; Line 23, P14 Line 3, P15; Line 23, P15 Line 22, P16. Second, a new figure (Figure 6) to show the overall national and regional trends for different periods and corresponding air pollution control policies. And we moved a table from supplementary materials to the main manuscript (see Table 2), which corresponds to Figure 6.

[Figure]

(6) P2, L3: polices-> policies

Response: We have corrected this mistake. See Line 3, P2.

(7) P3, L13-14: the citations are quite China-centric. Consider citing studies on SO2 and NO2 trends over China from non-Chinese groups, e.g. Itahashi et al. [2012], Krotkov et al. [2016], Miyazaki et al. [2017]. Suggested references: Krotkov, N. A., McLinden, C. A., Li, C., Lamsal, L. N., Celarier, E. A., Marchenko, S. V., Swartz, W. H., Bucsela, E. J., Joiner, J., Duncan, B. N., Boersma, K. F., Veefkind, J. P., Levelt, P. F., Fioletov, V. E., Dickerson, R. R., He, H., Lu, Z., and Streets, D. G.: Aura OMI observations of regional SO2 and NO2 pollution changes from 2005 to 2015, Atmos. Chem. Phys., 16, 4605-4629, doi:10.5194/acp-16-4605-2016, 2016. Miyazaki, K., Eskes, H., Sudo, K., Boersma, K. F., Bowman, K., and Kanaya, Y.: Decadal changes in global surface NOx emissions from multi-constituent satellite data assimilation, Atmos. Chem. Phys., 17, 807-837, https://doi.org/10.5194/acp-17-807-2017, 2017. Itahashi, S., Uno, I., Yumimoto, K., Irie, H., Osada, K., Ogata, K., Fukushima, H.,Wang, Z., and Ohara, T.: Interannual variation in the fine-mode MODIS aerosol optical depth and its relationship to the changes in sulfur dioxide emissions in China between 2000 and 2010, Atmos. Chem. Phys., 12, 2631-2640, https://doi.org/10.5194/acp-12-2631-2012, 2012.

Response: Thanks for the recommendation. These studies show that satellite remote sensing provides a powerful tool to assess the spatiotemporal trends of air pollutions for both global and regional scales. The references have been added in Lines 19-21, P3.

(8) P4, L11: pollution(s)

Response: This revision has been made (Line 15, P4).

(9) P4, L26: policy, not policies

Response: This revision has been made (Line 5, P5).

(10) P5, L21: unclear what R2 refers to

Response: It is coefficient of determination. We have added the description after R2 (L26, P5).

(11) P5, L25: "Validation results indicated…" be more specific. Validation done where, when?

Response: Two ways were used to validate the accuracy of historical estimates. First, we compared the historical estimates monitoring data from Hong Kong and Taiwan before 2013. Second, we estimated PM2.5 concentrations in the first half of 2014 using the 2013 model and compared them with the ground measurements to evaluate the accuracy of PM2.5 estimates beyond the model year, which can represent the accuracy of historical estimates. This description has been added (Lines 4-8, P6).

(12) P6, L12: suggest to remove referring to Ma et al. [2016]. This paper should describe the method briefly itself.

Response: We added details about the equations of the two-stage model, please see P8-P9.

(13) P6, L20-21: please discuss the representativeness of the PM2.5 stations for the size of a MODIS pixel, or vice versa?

Response: We pointed out the uneven spatial distribution of ground PM2.5 monitors. Please see Lines 6-12, P11.

(14) P9, L25: grammar

Response: This sentence has been deleted in our revision process.

(15) P10, L9-10: "strengthened the ECER measures" -> explain how

Response: Major air pollution control measures and corresponding achievements were added. See Lines 4-13, P13.

(16) P10, L11-12: explain qualitatively how this would have worked

Response: The main reasons were added. See Lines 22-27, P13.

(17) P10, L18-19: explain why further reduction emissions had no beneficial effect anymore

Response: The main reasons were added. See Lines 13-21, P14.

(18) P10, L21-22: rephrase... I don't think bottleneck is the term you should use.

Response: We have rephrase "bottleneck" to "limitation". See Line 13, P14.

(19) P10, L25: "After that" -> be more specific what the policy consisted of then

Response: Major measures included were added in Line 23, P14 Line 12, P15.

(20) P11, section 4.4: it would be useful to include here already how the findings relate to Chinese and WHO air quality standards.

Response: We thought that adding a comparison in Section 4.5 would be better. We added a sentence in Lines 10-12, P17. We want to show that although China has achieved great success in PM2.5 pollution control, PM2.5 levels are still much higher than Chinese and WHO air quality standards.

(21) P11, L6-7: how? we remain in the dark what was actually done and how that helped

Response: Major air pollution control measures and corresponding achievements were added. See Line 23, P15 Line 23, P16.

(22) P11, L11: what explains the regional differences?

Response: We discussed the regional differences in Lines 22-26, P19.

(23) P11, L21: close(d)

Response: This revision has been made. See Line 10, P17.

(24) P11, L25: what are the "official results"?

Response: They are the "official results" of "APPC-AP performance assessment (Table 4)". This revision has been made accordingly. See Line 13, P17,

(25) P12, L20: "the overall decrease" -> be quantitative

Response: Done. See Lines 23-24, P18; Lines 2, 7-8, P19.

(26) P13, L2: "All these policies" -> it should be made clear what was the essence of this

Response: Since we have added the essence in the main text, we did not add it here again. We refer it to Sections 4.4 and 4.5. See Lines 6-7, P19.

(27) P13, L4: MEE -> ?

Response: the Ministry of Ecology and Environment (MEE), see Line 3, P17.

(28) P13, L9: "air pollution control in China has achieved a periodic victory"-> this is awkward, do the authors mean that the measures taken so far have resulted in a temporary solution, or, more precisely, have succeeded to mitigate the worst aspects of PM2.5 pollution?

Response: What we want to say here is that air pollution control in China has achieved great success in PM2.5 pollution reduction. Sorry for the awkward phrase. We have revised this sentence to "Currently, China has achieved great success in PM2.5 pollution control." See Line 1, P20.

(29) Figure 2: unclear what difference is between upper and lower rows.

Response: They are model fitting (upper row) and cross validation (CV, lower row) results. We have revised the caption accordingly. We have revised the caption of Figure 2. And we added a brief description of CV in Line 25, P8 L3, P9.

Please also note the supplement to this comment:
https://www.atmos-chem-phys-discuss.net/acp-2018-1191/acp-2018-1191-AC1-supplement.zip
* * *
* * *
Interactive
comment

---

## Author Comment (AC2) · 31 Mar 2019

Responses to RC2

Please find the supplement for the revised manuscript and supplementary materials. We have highlighted the revisions in red font in the revised manuscript.

Comments from RC2:

The paper provides a useful overview of recent air quality control policies in China, while using an independent source of data to assess their efficacy. A statistical method is used to correlate satellite retrievals of Aerosol Optical Depth (AOD) to ground level

[Figure]

PM2.5 in China, by correlating AOD with meteorological data, fire spots and forest cover. It uses the large network of Chinese measurement stations to verify the model. The 2013 model, which was developed in another paper (Ma et al 2016) is used to project the concentration of PM2.5 backwards to 2005, while a separate model is developed each year for 2014 - 2017. This gives a 13-year PM2.5 dataset with complete spatial and temporal coverage for 2005 – 2017, which is then used to assess the success of China's air quality control policy that underwent significant changes during this period. Linear trends are calculated for the periods corresponding to specific policies (e.g. Five Year Plans). Calculated PM2.5 concentrations are also compared with official government data, to verify that targets were met. While this retrospective analysis of the success of China's control of PM2.5 pollution is very useful, the authors need to ensure that they acknowledge the role that inter-annual variation in meteorology may play in these trends.

Response: We would like to thank the reviewer for his valuable comments. We have revised the manuscript according to the comments, please see the following responses. For the impact of meteorological conditions, we have discussed this in Lines 11-14, 20-21, P19.

(1) Abstract The majority of the abstract summarizes the discussion section. A brief description of the two stage statistical model, including its predictors could be added.

Response: A brief description of the two stage statistical model and its predictors have been added in abstract. See Lines 8-11, P2.

(2) Intro P3, L23: It may be worth adding a sentence that briefly explains what the 'scaling method' is. There is a citation to Liu 2014 to back up the statement that, "Compared to the scaling method, statistical models have greater prediction accuracy but require large amount ground-measured PM2.5 data to develop the statistical models (Liu, 2014)". However, there is not a reference that corresponds to the "Liu, 2014" citation. Since the justification of method choice relies on this reference, it should be

added before the paper is reviewed again.

Response: Done. See Line 25, P3 Line 1, P4.

(3) Overview of air pollution control policies in China from 2005 to 2017 This section is a very broad summary of the actions within Five Year Plans and other major government directives that are relevant to air pollution control. The specific policies (e.g. 'Implement desulphurization and denitration facilities for coal-fired power sector and major industrial sectors') are summarised in Table 1, along with the metrics by which the policies' success will be judged. It may be useful to, where possible, cite government press releases/reports or literature that assess the success of these policies. However, the text in this section does not make any mention of the policies listen in Table 1. It would be useful for the reader for some information from Table 1 to be synthesised into this section, along with citations to previous studies that have attempted to assess the success of these policies (e.g. Schreifels et al, 2012) Reference: Schreifels, Jeremy J., Yale Fu, and Elizabeth J. Wilson. "Sulfur dioxide control in China: policy evolution during the 10th and 11th Five-year Plans and lessons for the future." Energy Policy48 (2012): 779-789.

Response: This comment is helpful. After careful consideration, we added major air pollution control measures, corresponding achievements, and how these policies were associated with PM2.5 pollutions in the main text and cited relevant references, including reference of Schreifels et al, 2012. See Lines 4-13, P13; Lines 22-27, P13; Lines 13-21, P14; Line 23, P14 Line 3, P15; Line 23, P15 Line 22, P16.

(4) P5, L13. It may be worth defining what China's 'new air quality standard' here, where it is first mentioned. It may be useful to provide the old air quality standard, and the name of the standard (GB 3095-2012). Currently the actual threshold number of China's air quality standard is first referenced of P13, L10 in the conclusion.

Response: Done. We briefly described the new air quality standard in Lines 4-11, P15.

(5) Data and Method P6, L19: Paper uses PM2.5 data from the CNEMC. Other papers, (e.g. Rohde and Muller (2015); Liu et al (2016)) have noted quality issues with this data. Were any quality control procedures applied to this data? References: Liu, Jianzheng, Weifeng Li, and Jie Li. "Quality screening for air quality monitoring data in China." Environmental pollution216 (2016): 720-723. Rohde, Robert A., and Richard A. Muller. "Air pollution in China: mapping of concentrations and sources." PloS one10.8 (2015): e0135749.

Response: Yes, we performed the data screening procedure before model fitting. Abnormal values (extreme high or extreme low values for a site compared with its neighboring sites, repeated values for continuous hours, etc.) were deleted before model fitting. We required at least 20 hourly records to calculate the daily average PM2.5 concentrations. Please see Lines 17-21, P7.

(6) Since the ground monitoring stations are typically within urban areas, could this bias the statistical model so that the PM2.5 predictions for non-urban areas is inaccurate? Why use the updated data to create separate statistical models for 2014, 2015, 2016 and 2017, yet only use the 2013 model to project back the PM2.5? Why should the 2013 model be more appropriate than the other years? Why not combine all the years where measurements are available? How is it justified to fit the model separately to the data in each province? Isn't using province boundaries somewhat arbitrary?

Response: Yes, we acknowledge this is a problem in statistical modeling of satellite PM2.5. We have discussed this in Lines 6-12, P11. There are two reasons that we only use the 2013 model to project back the PM2.5. First, the historical data were derived from our previous study, which only used the 2013 model. This dataset has been shown high accuracy and has been widely used in environmental epidemiological (Liu et al., 2016;Wang et al., 2018a), health impact (Liu et al., 2017;Wang et al., 2018b), and social economic impact (Chen and Jin, 2019;Yang and Zhang, 2018) studies in China. Second, a recent study has shown that the historical hindcast ability of the annual model decreased when hindcast year was long before the model year (Xiao et al.,

2018). Therefore, we did not use the models of 2014 to 2017 to estimate the hindcast PM2.5. For provincial models, we added the description how we fit the provincial model in Line 13-16, P8. We added the provincial results in Table S2-S4 (Supplementary Materials). And described the results in Line 23, P10 Line 5, P11. Results showed that the performance of first-stage LME model would greatly decreased if we fit the model for entire China.

(7) Many other studies of trends in atmospheric concentrations use a non-parametric trend estimator such as the Thiel-Sen slope estimator. The authors should justify their choice of the least squares regression to estimate the slope of the trend.

Response: In fact, the method we used in this study has been successfully applied to trend analyses of monthly mean PM2.5 and AOD anomaly time-series data (Hsu et al., 2012;Boys et al., 2014;Zhang and Reid, 2010;Xue et al., 2019). Therefore, we thought that the method we used is appropriate. See Lines 22-24, P9. Besides, we added a description of the method. Please see Lines 14-22, P9.

(8) In the results section, and Figures 6  7, a p threshold of 0.1 is mentioned, but you do not mention in the methods which statistical test you used to check the significance of your trends.

Response: The method of t test was used to obtain the statistical significance of the trends. See Lines 21-22, P9.

(9) Some of these questions about the methodology can be answered by reading the author's previous Ma et al 2016 paper, which is published in Environmental Health Perspectives. I recommend the authors reduce their reliance on referring to this previous paper, so that the methods section in the current paper can be understood without referring to another paper which the reader will not necessarily have access to.

Response: We added details about the equations of the two-stage model, please see P8-P9.

(10) P5, L26: Is it useful to the reader to list 9 studies that have referenced your previous paper? This list includes studies that this paper's co-authors are also co-authors on.

Response: These papers were the follow up studies using the PM2.5 dataset from 2004 to 2013 we developed in our previous study. Although some of them are the follow up studies by co-authors of this study, the publications of these studies show that the PM2.5 dataset has been widely recognized and used in academic field. And these references can support the rationality that we directly use this PM2.5 dataset from 2004 to 2013 in current study.

According to this comment, we have removed 3 references here (see Lines 13-15, P6) to simplify this paragraph.

(11) Results and Discussions Is it really useful to compare the PM2.5 trend with the corresponding FYP policies? This suggests that policies have immediate effects, and that they are the main contributor to the trends in PM2.5. There are other important confounding factors such as interannual variation in meteorology, China's economic output etc. May be best to avoid statements on the effectiveness of certain policies, or mention the above caveats in the conclusion.

Response: We added discussions about the impacts of meteorology and economic. See Lines 11-21, P19.

(12) I suggest the authors add a comparison of their results with other research that quantifies the trend in PM2.5 derived AOD in China, such as Lin et al., 2017. It may be interesting to perform a non-linear trend analysis on this dataset in certain key regions (e.g. Jing-Jin-Ji or PRD). Reference: Lin, C. Q., Liu, G., Lau, A. K. H., Li, Y., Li, C. C., Fung, J. C. H., Lao, X. Q. (2018). High-resolution satellite remote sensing of provincial PM2. 5 trends in China from 2001 to 2015. Atmospheric Environment, 180, 110-116.

Response: The revision has been made. We compared our results with two recent

studies. See Lines 3-13, P18.

(13) As you break down the trend into multiple overlapping periods of different lengths, it is difficult to get an overall impression of the rises and falls in the trend in different regions. Alternatively, a figure could be added with the yearly or monthly deseasonalised PM2.5 (averaged by different regions)

Response: We have added a new figure (Figure 6, P27) according to the comment. And we moved a table from supplementary materials to the main manuscript (see Table 2), which corresponds to Figure 6.

(14) I suggest the authors also mention the possibility of contribution of natural sources of aerosol to the trends. At P10, L16, the authors mention that the majority of the trend in PM2.5 during 2010-2013 are driven by decreases in Xinjiang and Central Inner Mongolia, which are both desert regions where the PM2.5 likely has a high dust component. This can be seen in your results. For example in panel (e) of Figure 7, where the western half of the Taklamakan desert has a strong positive trend, despite it being unlikely that there are large changes in emissions in this unpopulated region.

Response: The possible impact of dust in this region has been added. See Lines 6-9, P14.

(15) P3, L8: "However, the Chinese government did not realize the PM2.5 issues until 2012." This sentence seems disingenuous and qualitative so should be removed or rephrased.

Response: We have changed "realize" to "focus on". See Line 8, P3.

(16) P4, L6: Remove or replace the word 'preliminary'.

Response: We changed it to "preliminarily". Line 11, P4.

(17) P5 L14. "These policies indicated that the air pollution control in China began to focus on air quality improvement." This sentence could be rephrased, as it is currently

seems tautological.

Response: We changed it to "These policies indicated that the focus of air pollution control in China began to focus on PM2.5 concentrations reductions". See Lines 19-20, P5.

(18) P10, L22: The sentence "As the further development of social economic, the ECER policy had shown its bottleneck for PM2.5 reductions." does not make sense. Bottleneck may be the wrong word to describe this.

Response: We have rephrase "bottleneck" to "limitation". See Line 13, P14.

(19) P12, L25. Change 'to addressed' to "to address."

Response: Done. See Line 3, P19.

(20) P13, L6. 'Temporal' is not the right word here. Should be temporary?

Response: Done. See Line 13, P16.

References:

Boys, B., Martin, R., van Donkelaar, A., MacDonell, R., Hsu, C., Cooper, M., Yantosca, R., Lu, Z., Streets, D. G., Zhang, Q., and Wang, S.: Fifteen-year global time series of satellite-derived fine particulate matter, Environ. Sci. Technol., 48, 11109-11118, 2014.

Chen, S., and Jin, H.: Pricing for the clean air: Evidence from Chinese housing market, J. Clean. Prod., 206, 297-306, 2019.

Hsu, N. C., Gautam, R., Sayer, A. M., Bettenhausen, C., Li, C., Jeong, M. J., Tsay, S. C., and Holben, B. N.: Global and regional trends of aerosol optical depth over land and ocean using SeaWiFS measurements from 1997 to 2010, Atmos. Chem. Phys., 12, 8037-8053, 2012.

Liu, C., Yang, C., Zhao, Y., Ma, Z., Bi, J., Liu, Y., Meng, X., Wang, Y., Cai, J., and

Kan, H.: Associations between long-term exposure to ambient particulate air pollution and type 2 diabetes prevalence, blood glucose and glycosylated hemoglobin levels in China, Environ. Int., 92, 416-421, 2016.

Liu, M., Huang, Y., Ma, Z., Jin, Z., Liu, X., Wang, H., Liu, Y., Wang, J., Jantunen, M., Bi, J., and Kinney, P. L.: Spatial and temporal trends in the mortality burden of air pollution in China: 2004–2012, Environ. Int., 98, 75-81, 2017.

Wang, C., Xu, J., Yang, L., Xu, Y., Zhang, X., Bai, C., Kang, J., Ran, P., Shen, H., and Wen, F.: Prevalence and risk factors of chronic obstructive pulmonary disease in China (the China Pulmonary Health [CPH] study): a national cross-sectional study, The Lancet, 391, 1706-1717, 2018a.

Wang, Q., Wang, J., He, M. Z., Kinney, P. L., and Li, T.: A county-level estimate of PM2. 5 related chronic mortality risk in China based on multi-model exposure data, Environ. Int., 110, 105-112, 2018b.

Xiao, Q., Chang, H. H., Geng, G., and Liu, Y.: An ensemble machine-learning model to predict historical PM2. 5 concentrations in China from satellite data, Environ. Sci. Technol., 52, 13260-13269, 2018.

Xue, T., Zheng, Y., Tong, D., Zheng, B., Li, X., Zhu, T., and Zhang, Q.: Spatiotemporal continuous estimates of PM2. 5 concentrations in China, 2000–2016: A machine learning method with inputs from satellites, chemical transport model, and ground observations, Environ. Int., 123, 345-357, 2019.

Yang, J., and Zhang, B.: Air pollution and healthcare expenditure: Implication for the benefit of air pollution control in China, Environ. Int., 120, 443-455, 2018.

Zhang, J., and Reid, J. S.: A decadal regional and global trend analysis of the aerosol optical depth using a data-assimilation grade over-water MODIS and Level 2 MISR aerosol products, Atmos. Chem. Phys., 10, 10949-10963, 2010.

[Figure]

Please also note the supplement to this comment:
https://www.atmos-chem-phys-discuss.net/acp-2018-1191/acp-2018-1191-AC2-supplement.zip

---

## Author Response (AR2)

**Manuscript acp-2018-1191**

**Title: Effects of air pollution control policies on PM$_{2.5}$ pollution improvement in China from 2005 to 2017: a satellite based perspective**

Dear Editor,

We would like to thank you for your consideration of publication of this manuscript and your suggestion. According to your comment, we have compared our results with the recommended paper, which also reports trends in PM2.5 over China during the same period (Silver et al., 2018). Please see Lines 26-27, P16; Line 1, P17; and Lines 42-43, P22.

Besides, we added an affiliation for the first author (Please see the first page of the manuscript and Supplemental Material).

Should you have any questions, please feel free to contact me via email (njumazw@163.com).

Best regards,

Zongwei Ma, Ph.D.

Nanjing University